# Optimal Transport–Guided Stochastic Control for Graph Combinatorial Optimization

Yang Huang [1 2]   Yifan Zhang [1 2 3]   Jian Cheng [1 2 4]

## Abstract

We propose an OT-guided sampling framework for solving graph combinatorial optimization through exact multilinear relaxation. Graph combinatorial optimization problems can be written as quadratic unconstrained binary optimization(QUBO). Leveraging a classical result in combinatorial optimization, we obtain a continuous multi-linear relaxation of QUBO that is exact, in the sense that it preserves the optimal binary solutions. The challenge is that the resulting energy landscape is highly nonconvex. We address this by treating the objective as an energy function and optimizing via sampling from the induced Boltzmann distribution to escape poor local optima. Viewing sampling as transporting a simple reference distribution to the target distribution, we use optimal transport to characterize more efficient probability flow and derive a stochastic optimal control problem whose solution yields an optimal sampling dynamics. We parameterize the control policy with graph neural networks to approximate the optimal control. Experiments show improved solution quality and efficiency over strong combinatorial and learning-based baselines.

## 1. Introduction

Graph combinatorial optimization (GCO) problems such as Maximum Independent Set (MIS), Maximum Clique (MaxCl) and Maximum Cut (MaxCut) arise widely in network analysis, scheduling, and machine learning (Paschos, 2014; Nemhauser & Wolsey, 1988; Korte & Vygen, 2018; Papadimitriou & Steiglitz, 1998). The primary difficulty in

these problems lies in their discrete decision variables and strong coupling between variables, which together induce a highly rugged and combinatorial search space (Papadimitriou & Steiglitz, 1998). Many GCO problems can be naturally formulated as quadratic unconstrained binary optimization (QUBO), a canonical representation that captures these interactions in a unified form(Punnen, 2022). A classical result in combinatorial optimization shows that QUBO admits an exact continuous multi-linear relaxation: the continuous problem has no relaxation gap and admits an optimal binary solution (Merz & Freisleben, 2002; Zemel, 1981). By lifting discrete variables to a continuous domain, this relaxation enables the use of gradient information and continuous optimization tools without introducing relaxation gaps. Compared to convex or semidefinite relaxations(Shor, 1987; Poljak et al., 1995; Billionnet et al., 2009), exact multi-linear relaxation retains solution fidelity and offer a principled bridge between combinatorial structure and continuous optimization.

However, the benefits of multi-linear relaxation come with a notable trade-off. While continuity allows local gradient information to guide optimization, the resulting objective function becomes highly nonconvex, inheriting complex interactions from the original combinatorial problem. The resulting energy landscape typically contains numerous poor local optima, saddle points, and flat regions, posing challenges for effective exploration and optimization (Hillar & Lim, 2013; Ryoo & Sahinidis, 2001). As a consequence, standard deterministic optimization methods, including gradient-based approaches (Gill & Murray, 1972; Kingma & Ba, 2015), may struggle to adequately explore the landscape and become trapped in suboptimal stationary points. A common strategy for tackling difficult continuous optimization problems is to interpret the objective function as an energy function and induce a target distribution that concentrates probability mass on low-energy regions (Geman & Geman, 1984). Sampling from such energy-based distributions provides a principled mechanism for optimization, as it naturally biases the search toward high-quality solutions while retaining the ability to explore beyond local neighborhoods. This perspective has been widely adopted in nonconvex optimization to improve exploration and reduce entrapment in poor local optima (Geman & Hwang, 1986; Ma et al., 2019).

[1]C²DL, Institute of Automation, Chinese Academy of Sciences, Beijing, China [2]School of Artificial Intelligence, University of Chinese Academy of Sciences, Beijing, China [3]University of Chinese Academy of Sciences, Nanjing, China [4]AiRiA, Nanjing, China. Correspondence to: Yifan Zhang <yfzhang@nlpr.ia.ac.cn>.

*Proceedings of the 43rd International Conference on Machine Learning*, Seoul, South Korea. PMLR 306, 2026. Copyright 2026 by the author(s).

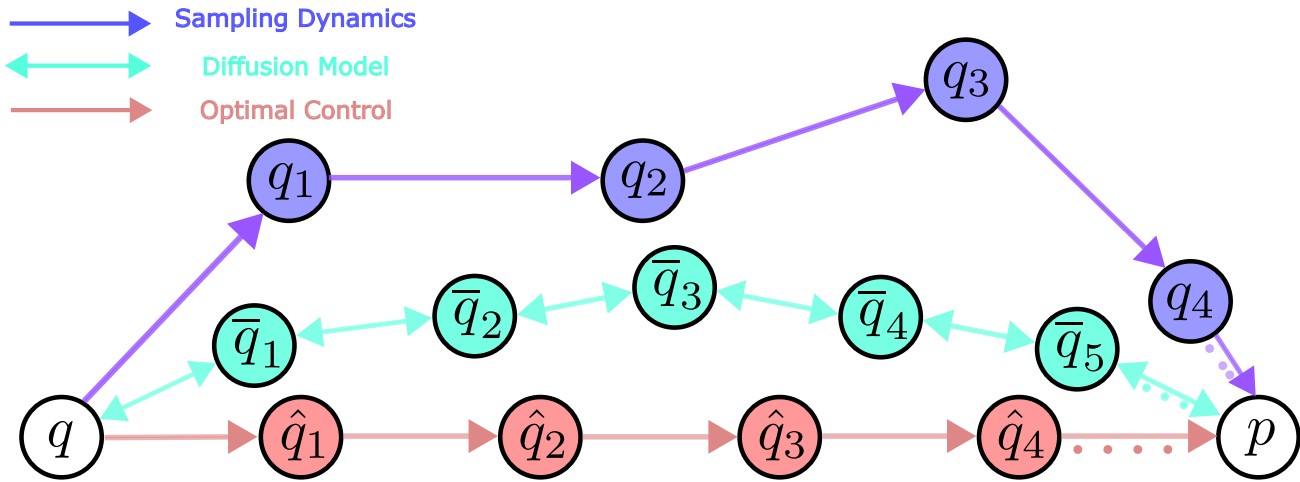

*Figure 1.* Illustration of sampling-based optimization from a probabilistic and geometric perspective. The reference distribution and the target energy-induced distribution are represented as points in probability space. Different sampling methods correspond to different trajectories connecting these distributions. Generic sampling dynamics (purple) and diffusion-based methods (blue) may follow indirect paths, while optimal transport characterizes more efficient trajectories (pink) that motivate the design of sampling dynamics in this work.

Our approach is motivated by a probabilistic and geometric viewpoint on sampling-based optimization. From this viewpoint, sampling can be interpreted as transforming a simple reference distribution into a target distribution induced by the energy function. Different sampling procedures (Sun et al., 2023; Feng & Yang, 2025; Sun et al., 2021; Grathwohl et al., 2021) correspond to different probability flow trajectories connecting reference distribution and target distribution. Many generic sampling dynamics, including diffusion-based methods (Austin et al., 2021; Sun & Yang, 2023; Sanokowski et al., 2025) used for combinatorial optimization, may follow indirect trajectories in probability space, which may result in slower progress under a fixed compute budget. Optimal transport (Chewi et al., 2024) provides a principled framework for reasoning about efficient probabilistic transformations by characterizing paths that minimize transportation cost between distributions. Figure 1 illustrates this geometric intuition.

In particular, entropic optimal transport and its dynamic Benamou–Brenier representation (Benamou & Brenier, 2000) admit equivalent stochastic-control (Schrödinger bridge) formulations. Leveraging this connection, we obtain a stochastic optimal control view of sampling the target distribution induced by the multilinear relaxation of QUBO, where the drift is optimized to guide the process toward low-energy regions efficiently. Although the optimal control admits an analytical characterization, directly leveraging this optimal control to perform efficient sampling remains highly challenging in practice. To address this issue, we propose a practical paradigm that combines a Laplace approximation with a graph neural network parameterization of the control policy, enabling scalable and structure-aware sampling dynamics. Our contributions are summarized as follows:

- **OT-guided sampling framework for exact continuous QUBO formulations.** We optimize the exact multi-linear continuous formulation via energy-based sampling, providing a gap-free continuous route to high-quality discrete solutions despite severe nonconvexity.

- **Practical and scalable approximation of the induced control.** We combine a Laplace approximation with a GNN-parameterized control policy to approximate the optimal drift, yielding structure-aware sampling dynamics that scale to large graphs.

- **Strong empirical performance on standard benchmarks.** On MIS, MaxCl and MaxCut, our method improves solution quality over strong learning-based and sampling-based baselines in our experiments.

## 2. Related Work

Several recent works have explored sampling-based approaches for combinatorial optimization by designing local proposal mechanisms guided by objective information. For example, GWG (Grathwohl et al., 2021) uses a first-order approximation to estimate energy changes within the 1-Hamming-ball neighborhood, steering each proposal toward high-density regions. However, the reliance on small local neighborhoods often leads to poor exploration and local-optima issues. To alleviate this limitation, PAS/PAFS (Sun et al., 2021) propose sequentially sampling multiple coordinates at each step, thereby enlarging the effective neighborhood explored by the sampler. To address the inefficiency of

sequential sampling, (Zhang et al., 2022) derive a discrete Langevin sampler by mapping continuous Langevin dynamics to the discrete domain, enabling parallel updates across all coordinates with gradient-informed transition probabilities. However, directly adapting such discrete Langevin dynamics to a simulated annealing framework overlooks key differences between continuous and discrete spaces, often resulting in severe local-optima trapping. More recently, RLSA (Feng & Yang, 2025) constrains the number of variables updated at each step to improve stability and efficiency. In contrast to these approaches, our work builds on discrete MCMC developments by proposing a simple framework that unifies parallel sampling with effective mechanisms for escaping poor local optima. More related works about learning-based method are provided in Appendix A.

## 3. Problem Formulation and Exact Multi-Linear Relaxation

### 3.1. Quadratic Unconstrained Binary Optimization

We consider graph combinatorial optimization problems that can be formulated as *quadratic unconstrained binary optimization (QUBO)*. Let $\boldsymbol{Q} \in \mathbb{R}^{n \times n}$ and $\boldsymbol{c} \in \mathbb{R}^n$, QUBO is defined as the mathematical programming:

$$\min_{\boldsymbol{x} \in \{0,1\}^n} \quad E(\boldsymbol{x}) \triangleq \boldsymbol{x}^\top \boldsymbol{Q} \boldsymbol{x} + \boldsymbol{c}^\top \boldsymbol{x}, \qquad (1)$$

This formulation serves as a canonical representation for a wide range of classical graph combinatorial optimization problems, including Maximum Independent Set, Maximum Cut and Maximum Clique. For completeness, we provide standard reductions from these problems to QUBO in Appendix B.1.

### 3.2. Exact Multi-linear Relaxation

Consider a QUBO objective induced by a graph $G = (V, E)$ with $|V| = n$ nodes:

$$E(\boldsymbol{x}; G) = \boldsymbol{x}^\top \boldsymbol{Q} \boldsymbol{x} + \boldsymbol{c}^\top \boldsymbol{x}, \quad \boldsymbol{x} \in \{0,1\}^n, \qquad (2)$$

where we assume $\boldsymbol{Q}$ is symmetric (otherwise we replace $\boldsymbol{Q}$ by $(\boldsymbol{Q}+\boldsymbol{Q}^\top)/2$ without changing the objective). Expanding the quadratic form yields

$$E(\boldsymbol{x}; G) = \sum_{i=1}^n (Q_{ii} + c_i) x_i + \sum_{1 \le i < j \le n} 2 Q_{ij} x_i x_j, \quad (3)$$

where we used $x_i^2 = x_i$ for binary variables. The first sum is linear, and the second sum is multilinear (in fact, bilinear, since it only involves pairwise products).

The *multi-linear relaxation* is obtained by lifting the domain to $\boldsymbol{x} \in [0,1]^n$ while keeping the same linear and bilinear structure:

$$f(\boldsymbol{x}; G) \triangleq \sum_{i=1}^n (c_i + Q_{ii}) x_i + \sum_{1 \le i < j \le n} 2 Q_{ij} x_i x_j, \ \boldsymbol{x} \in [0,1]^n. \qquad (4)$$

Equivalently, $f$ can be written as

$$f(\boldsymbol{x}; G) = \mathbb{E}_{\boldsymbol{z} \sim \prod_i \mathrm{Bern}(x_i)} \left[ \boldsymbol{z}^\top \boldsymbol{Q} \boldsymbol{z} + \boldsymbol{c}^\top \boldsymbol{z} \right],$$

which makes clear that $f$ is multi-linear and satisfies $f(\boldsymbol{x}; G) = E(\boldsymbol{x}; G)$ for all $\boldsymbol{x} \in \{0,1\}^n$. The resulting exact multilinear relaxation is

$$\min_{\boldsymbol{x} \in [0,1]^n} \quad f(\boldsymbol{x}; G), \qquad (5)$$

which introduces no relaxation gap in optimal value and admits an optimal binary minimizer.

### 3.3. Exactness Property

The key property of the multi-linear relaxation is its exactness, which we state below:

**Proposition 3.1.** *Given graph $G = (V, E)$ with $|V| = n$ nodes and problem type, let $E(\cdot; G) : \{0,1\}^n \to \mathbb{R}$ be the QUBO objective in (2), and let $f(\cdot; G) : [0,1]^n \to \mathbb{R}$ denote its multilinear relaxation defined in (4). Then, there exists $\boldsymbol{x}^\star \in \{0,1\}^n$ such that*

$$f(\boldsymbol{x}^\star; G) = \min_{\boldsymbol{x} \in [0,1]^n} f(\boldsymbol{x}; G) = \min_{\boldsymbol{x} \in \{0,1\}^n} E(\boldsymbol{x}; G).$$

The multilinear relaxation is exact in optimal value: it introduces no relaxation gap, because the continuous problem admits an optimal solution at a binary point $\boldsymbol{x}^\star \in \{0,1\}^n$. However, the multilinear relaxation may admit fractional minimizers, i.e., integrality of all minimizers is not guaranteed. A simple proof of Proposition 3.1 can be found in Appendix B.2.

## 4. Sampling-Based Optimization via Entropic Optimal Transport and Stochastic Control

### 4.1. Energy-Based Sampling and Efficient Search Paths

Given the exact multilinear relaxation in Section 3, the objective $f(\boldsymbol{x}; G)$ defines a highly nonconvex energy landscape. To enable effective exploration beyond poor local optima, we adopt an energy-based sampling perspective and define the target distribution:

$$\pi^\star(\boldsymbol{x}; G) \propto \exp(-f(\boldsymbol{x}; G)), \qquad (6)$$

whose modes correspond to low-energy, high-quality solutions.

From this viewpoint, sampling-based optimization can be interpreted as transforming an easy-to-sample reference distribution $\rho_0$ into the target distribution $\pi^\star$. Different sampling algorithms correspond to different ways of transporting probability mass between distributions. This naturally raises the question of how to characterize *efficient* stochastic search paths between $\rho_0$ and $\pi^\star$.

Entropic optimal transport (OT) provides a principled notion of efficient stochastic transport between distributions. Given two distributions $\rho_0$ and $\rho_1$, static entropic OT (Chewi et al., 2024) is defined as

$$\min_{\pi \in \Pi(\rho_0, \rho_1)} \left\{ \int c(x, y)\, \pi(x, y)\, dx\, dy \;+\; \varepsilon\, H[\pi(x, y)] \right\},$$
(7)

where $\Pi(\rho_0, \rho_1)$ denotes the set of joint distributions with marginals $\rho_0$ and $\rho_1$, $c(x, y)$ is a transportation cost measuring the effort of moving probability mass from $x$ to $y$, and $\varepsilon > 0$ controls the strength of the entropy regularization term. In this work we use the quadratic cost $c(x, y) = \|x - y\|^2$ and its Schrödinger-bridge (Brownian reference) dynamic representation.

The entropy term encourages stochasticity and exploration, while the transport cost promotes efficiency. As a result, entropic OT defines an optimal trade-off between efficient transport and random exploration, making it a natural criterion for assessing the quality of stochastic search paths. In our setting, we consider $\rho_1 = \pi^\star$, so that entropic OT characterizes the most efficient stochastic transport from a reference distribution to the target energy-induced distribution.

### 4.2. OT-Guided Stochastic Optimal Control

The static entropic optimal transport formulation in Eq. (7) operates at the distributional level and does not directly yield sample trajectories, which limits its applicability for sampling-based optimization. Our use of OT is therefore structural rather than a claim that the final algorithm exactly solves a standard OT/SB problem: the quadratic control cost and controlled diffusion define an OT/SB-inspired search dynamics, while the terminal objective is adapted to concentrate samples on low-energy solutions instead of matching the full target distribution. To obtain an implementable sampling procedure, prior work has shown that entropic OT admits equivalent dynamic and particle-based representations, notably through the Schrödinger bridge and stochastic optimal control formulations (Benamou & Brenier, 2000; Léonard, 2014).

Following this line of work, consider a controlled stochastic process $\{X_t\}_{t \in [0,1]}$ governed by

$$dX_t = u_t(X_t)\, dt + \sqrt{2\varepsilon}\, dW_t,$$
(8)

where $u_t$ denotes a time-dependent drift function, $\varepsilon > 0$ controls the noise magnitude, and $W_t$ is standard Brownian motion. Let $\rho_t$ denote the marginal distribution of $X_t$. Under this particle-based formulation, the entropic OT objective can be realized by the stochastic optimal control problem:

$$\min_{u} \int_0^1 \mathbb{E}_{X_t \sim \rho_t} \left[ \frac{1}{4\varepsilon} \|u_t(X_t)\|^2 \right] dt + \lambda_h \mathrm{CE}[\rho_1 \| \pi^\star],$$
(9)

where $\mathrm{CE}[\rho_1 \| \pi^\star]$ denotes the cross entropy of the terminal distribution $\rho_1$ relative to the target distribution $\pi^\star$ and $\lambda_h > 0$ is a weighting parameter. This formulation provides a practical mechanism for generating samples that transport probability mass efficiently toward the target distribution $\pi^\star$.

### 4.3. Value Function and Optimal Control

The stochastic optimal control problem in (9) admits a value function that characterizes the optimal sampling dynamics.

**Proposition 4.1.** *Let $V(t, x; G)$ denote the value function associated with the SOC problem in (9). Let $\Phi(x; G) = \lambda_h f(x; G)$, then $V$ admits the representation:*

$$V(t, x; G) = -\log\left( \int_{[0,1]^n} p_{1-t}(x, y)\, e^{-\Phi(y; G)}\, dy \right),$$
(10)

*where the heat kernel is*

$$p_s(x, y) \;=\; (4\pi\varepsilon s)^{-n/2} \exp\left( -\frac{\|x - y\|^2}{4\varepsilon s} \right).$$

*Then the optimal control (drift function) admits the form:*

$$u_t^\star(x; G) = -2\varepsilon \nabla_x V(t, x; G) = \frac{\mathbb{E}_{Y \sim q_{t,x}}[Y] - x}{1 - t},$$
(11)

$$q_{t,x}(y) \propto p_{1-t}(x, y)\, e^{-\Phi(y; G)},$$
(12)

*which induces sampling dynamics that efficiently transport probability mass toward regions of low energy $f(x; G)$.*

The derivation of the value function representation in Appendix C.

### 4.4. Laplace Approximation and Amortized GNN Computation

Equation (11) provides an exact expression of the optimal control in terms of the conditional expectation. Despite its closed form, directly evaluating this expectation is computationally prohibitive in high dimensions, as it requires integration over the continuous domain $[0, 1]^n$ for each time–state pair $(t, x)$.

**Algorithm 1** Amortized GNN Training with Rollout Buffer

**Require:** Laplace energy $L(\cdot; G)$; graph $G$; truncation $\delta > 0$; iterations $N_{\text{iter}}$; particles $K$; sampled time points per rollout $T$; batch size $B$
1: Initialize GNN parameters $\theta$ and replay buffer $\mathcal{B} \leftarrow \emptyset$
2: **for** $i = 1$ to $N_{\text{iter}}$ **do**
3:    **(Collect)** Sample $K$ initial states $\{\boldsymbol{x}_0^{(j)}\}_{j=1}^K \sim \rho_{\text{ref}}$
4:    Sample $\{\tilde{t}_k\}_{k=0}^T$ i.i.d. from $\text{Unif}[0, 1-\delta]$, and sort: $0 \leq t_0 \leq t_1 \leq \cdots \leq t_T \leq 1-\delta$
5:    **for** $k = 0$ to $T - 1$ **do**
6:       Set $\Delta t_k = t_{k+1} - t_k$
7:       **for** $j = 1$ to $K$ **(in parallel) do**
8:          Compute $\boldsymbol{y}_k^{(j)} = \boldsymbol{y}_\theta(t_k, \boldsymbol{x}_k^{(j)}; G)$
9:          Compute control $\hat{\boldsymbol{u}}_k^{(j)}$ using (17)
10:         Sample $\boldsymbol{\xi}_k^{(j)} \sim \mathcal{N}(\boldsymbol{0}, \boldsymbol{I})$
11:         Update and project:

$$\boldsymbol{x}_{k+1}^{(j)} = \Pi_{[0,1]^n}\left(\boldsymbol{x}_k^{(j)} + \hat{\boldsymbol{u}}_k^{(j)}\,\Delta t_k + \sqrt{2\varepsilon \Delta t_k}\,\boldsymbol{\xi}_k^{(j)}\right)$$

12:         Store $(t_k, \boldsymbol{x}_k^{(j)})$ in buffer $\mathcal{B}$ (discard oldest if full)
13:       **end for**
14:    **end for**
15:    **(Update)** Sample $\{(t^{(b)}, \boldsymbol{x}^{(b)})\}_{b=1}^B$ uniformly from $\mathcal{B}$
16:    Compute $\boldsymbol{y}_\theta^{(b)} = \boldsymbol{y}_\theta(t^{(b)}, \boldsymbol{x}^{(b)}; G)$
17:    Update $\theta$ by minimizing (16) on the mini-batch
18: **end for**
19: **return** $\theta$

---

To obtain a tractable approximation, we exploit the concentration property of the heat kernel in the small-noise regime. Specifically, define the Laplace energy

$$L(t, \boldsymbol{x}, \boldsymbol{y}; G) = \Phi(\boldsymbol{y}; G) + \frac{\|\boldsymbol{x} - \boldsymbol{y}\|^2}{4\varepsilon(1 - t)}. \tag{13}$$

and let

$$\boldsymbol{y}^\star(t, \boldsymbol{x}; G) = \underset{\boldsymbol{y} \in [0,1]^n}{\arg\min}\, L(t, \boldsymbol{x}, \boldsymbol{y}; G). \tag{14}$$

When $\varepsilon(1 - t)$ is small, a first-order Laplace approximation yields

$$\mathbb{E}_{\boldsymbol{Y} \sim q_{t,\boldsymbol{x}}}[\boldsymbol{Y}] \approx \boldsymbol{y}^\star(t, \boldsymbol{x}; G),$$

leading to the computable surrogate control:

$$\hat{\boldsymbol{u}}(t, \boldsymbol{x}; G) = -\frac{\boldsymbol{x} - \boldsymbol{y}^\star(t, \boldsymbol{x}; G)}{1 - t}. \tag{15}$$

The minimization problem in (14) is closely related to a proximal map associated with the energy $H$. A brief justification of this approximation and the associated assumptions are provided in Appendix D.

**Algorithm 2** Inference with Amortized Control and Multiple Particles

**Require:** Trained GNN $\boldsymbol{y}_\theta$; graph $G$; particles $K$; steps $T$; noise level $\varepsilon$; truncation $\delta > 0$; initial states $\{\boldsymbol{x}_0^{(j)}\}_{j=1}^K$
1: Set $\Delta t = (1 - \delta)/T$ and $t_k = k\Delta t$
2: **for** $k = 0$ to $T - 1$ **do**
3:    **for** $j = 1$ to $K$ **(in parallel) do**
4:       $\boldsymbol{y}_k^{(j)} = \boldsymbol{y}_\theta(t_k, \boldsymbol{x}_k^{(j)}; G)$
5:       $\hat{\boldsymbol{u}}_k^{(j)}$ by (17)
6:       Sample $\boldsymbol{\xi}_k^{(j)} \sim \mathcal{N}(\boldsymbol{0}, \boldsymbol{I})$
7:       Update and project:

$$\boldsymbol{x}_{k+1}^{(j)} = \Pi_{[0,1]^n}\left(\boldsymbol{x}_k^{(j)} + \hat{\boldsymbol{u}}_k^{(j)}\,\Delta t + \sqrt{2\varepsilon \Delta t}\,\boldsymbol{\xi}_k^{(j)}\right)$$

8:    **end for**
9: **end for**
10: **return** $\{\boldsymbol{x}_k^{(j)}\}_{k=0, j=1}^{T, K}$

---

Solving (14) repeatedly by iterative optimization is still costly (see Section 5.3). We therefore amortize the Laplace subproblem with a GNN $\boldsymbol{y}_\theta(t, \boldsymbol{x}; G)$ and train it using rollout data. For each rollout, we initialize particles by sampling $\boldsymbol{x}_0 \sim \rho_0 = \text{Unif}([0, 1]^n)$, and sample $T$ time points $\{\tilde{t}_k\}_{k=0}^T$ i.i.d. from $\text{Unif}[0, 1-\delta]$, then sort them to obtain $0 \leq t_0 \leq \cdots \leq t_T \leq 1-\delta$ and simulate using $\Delta t_k = t_{k+1} - t_k$. This randomized discretization exposes the model to varying step sizes and improves generalization across different $\Delta t$ at test time, while $\delta > 0$ avoids numerical issues from the $(1 - t)^{-1}$ factor.

We store rollout samples $(t, \boldsymbol{x})$ in a replay buffer and train $\boldsymbol{y}_\theta$ by minimizing

$$\min_\theta \mathbb{E}_{t \sim \mu,\, \boldsymbol{x} \sim \rho_t}\left[L\big(t, \boldsymbol{x}, \boldsymbol{y}_\theta(t, \boldsymbol{x}; G); G\big)\right], \tag{16}$$

where $\mu$ is supported on $[0, 1 - \delta]$ and $\rho_t$ denotes the time-marginal distribution induced by the current dynamics (approximated by the buffer). During each update, the buffer samples are treated as a fixed data distribution, so gradients are not back-propagated through $\rho_t$ or the rollout collection process. This detached training objective avoids the degenerate behavior we observed when directly minimizing (9), where the learned control collapses to nearly zero except near the terminal time. After training, we replace $\boldsymbol{y}^\star$ with $\boldsymbol{y}_\theta$ in Eq. (15) to obtain the amortized control

$$\hat{\boldsymbol{u}}_\theta(t, \boldsymbol{x}; G) = -\frac{\boldsymbol{x} - \boldsymbol{y}_\theta(t, \boldsymbol{x}; G)}{1 - t}, \tag{17}$$

and run multiple particles in parallel for inference (Algorithm 2). The GNN uses a sigmoid output so that $\boldsymbol{y}_\theta \in [0, 1]^n$, and after each stochastic update we project the state back to $[0, 1]^n$. The complete training and infer-

ence procedures are summarized in Algorithms 1 and 2, respectively.

We interpret the final relaxed solution $\boldsymbol{x} \in [0, 1]^n$ as selection probabilities. To obtain a feasible integer solution, we apply a simple greedy rounding procedure: we sort nodes in decreasing order of $x_u$, and iteratively add node $u$ into the selected set $S$ if and only if $S \cup \{u\}$ satisfies the problem constraints. The resulting set $S$ is therefore feasible by construction. In practice the relaxed states are typically close to binary; simple thresholding with feasibility repair gave slightly worse results in our preliminary tests.

# 5. Experiments

This section evaluates the proposed method as a solver for graph combinatorial optimization problems. We focus on Maximum Independent Set (MIS), Maximum Cut (MaxCut) and Maximum Clique (MaxCl), which are representative NP-hard problems with highly nonconvex energy landscapes.

## 5.1. Experimental Setup

**Tasks.** We consider three classical graph combinatorial optimization problems: Maximum Independent Set (MIS), Maximum Clique (MaxCl) and MaxCut. All problems are formulated as QUBO and solved via the proposed multilinear relaxation and sampling-based optimization framework. These three problems cover both constrained and unconstrained optimization settings. MaxCut is an unconstrained quadratic optimization problem defined solely by pairwise interactions, while MIS and Maximum Clique are constrained combinatorial problem that enforces hard feasibility constraints. Moreover, Maximum Clique can be reduced to MIS through standard polynomial-time transformations. The transformations and additional experiments on these related problems are provided in Appendix E.

**Graph Instances.** We evaluate the proposed method on synthetic graph instances generated using standard models from the combinatorial optimization literature.

For Maximum Independent Set and Maximum Clique, we primarily evaluate on instances generated by the RB model (Xu et al., 2005), which is known to yield challenging random instances. We consider two graph size ranges, with the number of vertices in 200–300 (small) and 800–1200 (large), respectively. In addition, to assess robustness on more generic random graphs, we further evaluate MIS on Erdős–Rényi (ER) graphs (Erdős & Rényi, 1959) with the number of nodes sampled from 700–800. This provides a complementary benchmark with a more homogeneous random structure.

For Maximum Cut, we use graphs generated by the Barabási–

Albert (BA) model (Barabási & Albert, 1999), which produces scale-free networks with power-law degree distributions and hub structures commonly observed in real-world graphs. Such structural properties can pose additional challenges for optimization algorithms. We evaluate two graph size ranges, with the number of vertices in 200–300 (small) and 800–1200 (large), respectively.

The more detailed instance generation procedures are provided in Appendix E.2.

**Baselines.** We compare with representative baselines commonly used for graph combinatorial optimization, applied to each task when applicable (see Tables 1 and 2). **Optimization-based (OR)** baselines include ILP formulations solved by Gurobi (Gurobi Optimization, LLC, 2024) for MIS, MaxCl, and MaxCut, and KaMIS (Lamm et al., 2017) as a strong problem-specific MIS solver. **Heuristic / sampling-based (H)** baselines include Greedy and Mean-Field Annealing (MFA) (reported for MaxCl/MaxCut; omitted on MIS due to consistently inferior performance) (Bilbro et al., 1988). **Supervised learning (SL)** baselines include DGL (Böther et al., 2022) (MCTS+GNN) and DIFUSCO (Sun & Yang, 2023) (diffusion-based) for MIS. **Unsupervised learning (UL)** baselines include LTFT (Zhang et al., 2023) (GFlowNet/GIN) as well as diffusion-style approaches DiffUCO (Sanokowski et al., 2024) and SDDS (Sanokowski et al., 2025) (applied to MIS/MaxCut when available), and ERDOES (Erdős & Rényi, 1960) for MaxCut and MaxCl. **Reinforcement learning (RL)** baselines include PPO-based methods (Ahn et al., 2020), DIMES (Qiu et al., 2022), and RLNN (Feng & Yang, 2025) for MIS and, when applicable, MaxCut.

**Evaluation Protocol.** For our methods, models are trained on a set of 5,000 graph instances. Evaluation is conducted on held-out test sets, and all reported results are averaged over the test instances. For the RB and BA datasets (across all graph sizes), we use 500 test instances. For the ER datasets, we evaluate on a test set of 128 instances, following common practice for this benchmark. For our method, the notation $T \times K$ denotes $T$ optimization steps with $K$ particles. In particular, $T = 500$ corresponds to $\Delta t = 0.002$ with $K = 64$ particles, and $T = 100$ corresponds to $\Delta t = 0.01$ with $K = 20$ particles. We rerun PPO and RLNN under our evaluation protocol; other baseline numbers are taken from the corresponding papers when available. Cross-scale transfer results are reported in Appendix E.4. More information about experiments setting including network architecture and hyperparameters are provided in Appendix E.3

## 5.2. Main Results: Solution Quality

Tables 1 and 2 report the solution quality of all compared methods on the MIS, MaxCl, and MaxCut benchmarks.

*Table 1.* Comparison of solution quality and runtime for the MIS problem. Results are reported on **RB-[200–300]**, **RB-[800–1200]**, and **ER-[700–800]**. For each setting, we report the average independent set size (higher is better) and the *average wall-clock time per test instance* (seconds; lower is better), computed under each method's default inference setting; for our method we report two computational budgets ($T \times K$). For our method, the notation $T \times K$ denotes $T$ optimization steps with $K$ particles. In particular, $T = 500$ corresponds to $\Delta t = 0.002$ with $K = 64$ particles, and $T = 100$ corresponds to $\Delta t = 0.01$ with $K = 20$ particles. A dash (–) indicates that a method is not applicable or could not be run under the given setting. The best result among learning-based methods in each column is highlighted in bold.

| MIS | | RB-[200–300] | | RB-[800–1200] | | ER-[700–800] | |
|---|---|---|---|---|---|---|---|
| **Method** | **Type** | Size ↑ | Time ↓ | Size ↑ | Time ↓ | Size ↑ | Time ↓ |
| Gurobi | OR | 20.21 | 0.41 | 40.90 | 15.624 | 41.38 | 6.000 |
| KaMIS | OR | 20.23 | 10.080 | 43.15 | 14.760 | 44.87 | 6.256 |
| Convex-Relax(0.15, 1, 200) | OR | 18.13 | 0.090 | 34.42 | 1.330 | 36.26 | 1.460 |
| Decoder(0, 1) | OR | 15.73 | 0.004 | 31.88 | 0.070 | 30.50 | 0.060 |
| Convex-Relax(0, 1, 1) | OR | 17.01 | 0.060 | 33.00 | 1.210 | 33.94 | 1.340 |
| Bilinear-Relax(0, 1, 16) | OR | 17.31 | 37.212 | – | – | – | – |
| DGL | SL | 17.36 | 1.534 | 34.50 | 2.868 | 37.26 | 2.725 |
| DIFUSCO | SL | 18.52 | 1.926 | – | – | 41.12 | 3.200 |
| LTFT | UL | 19.18 | 0.064 | 37.48 | 0.524 | 41.07 | 2.21 |
| DiffUCO | UL | 19.59 | 0.056 | 38.87 | 5.924 | 43.32 | 0.82 |
| SDDS | UL | 19.62 | 0.040 | **39.32** | 5.762 | – | – |
| PPO | RL | 18.43 | 0.154 | 32.32 | 0.906 | – | – |
| DIMES | RL | – | – | – | – | 42.06 | 1.441 |
| RLNN | RL | 19.52 | 0.197 | 38.06 | 4.325 | 43.34 | 0.164 |
| Ours(500×64) | UL | **20.12** | 6.381 | 39.25 | 9.93 | **43.41** | 7.934 |
| Ours(100×20) | UL | 19.652 | 0.562 | 38.12 | 4.738 | 42.09 | 1.981 |

Overall, our method achieves strong and competitive performance across different problem settings, while maintaining consistent behavior across graph distributions and scales.

**Maximum Independent Set.** Table 1 presents results on the MIS problem over RB and ER graph families. On the **RB-[200–300]** and **ER-[700–800]** datasets, our method with a larger computational budget (Ours $500 \times 64$) achieves the best average independent set size among all learning-based compared approaches. On the more challenging **RB-[800–1200]** setting, our method remains highly competitive, closely matching the performance of the strongest unsupervised baseline (SDDS) and optimization-based solvers. Compared with supervised learning methods such as DGL and DIFUSCO, our approach consistently yields larger independent sets across all applicable settings. The smaller configuration (Ours $100 \times 20$) further demonstrates that competitive solution quality can be preserved with substantially reduced computational cost.

**Maximum Clique.** The left half of Table 2 summarizes results on the MaxCl problem. Across both **RB-[200–300]** and **RB-[800–1200]**, our method achieves the largest or near-largest clique sizes, outperforming heuristic baselines and

most learning-based methods. In particular, the configuration with a larger computational budget consistently matches or exceeds the performance of strong unsupervised baselines, while remaining competitive with the optimization-based solver as the problem size increases.

**Maximum Cut.** The right half of Table 2 reports results on the MaxCut problem over Barabási–Albert graphs. Our method consistently achieves the highest cut values on both **BA-[200–300]** and **BA-[800–1200]**. The advantage becomes more pronounced on larger graphs, where our approach outperforms unsupervised generative methods, diffusion-based approaches, and reinforcement learning baselines. These results demonstrate the strong scalability of our method and its effectiveness on structurally heterogeneous graphs.

Overall, the results across all tasks indicate that our method provides a strong balance between solution quality and computational flexibility, making it applicable to a wide range of graph combinatorial optimization problems.

*Table 2.* Comparison of solution quality and normalized runtime for the MaxCl and MaxCut problems. The left half of the table reports results for the MaxCl problem on **RB-[200–300]** and **RB-[800–1200]**, while the right half reports results for the MaxCut problem on Barabási–Albert (BA) graphs, including **BA-[200–300]** and **BA-[800–1200]**. For both tasks, we report the solution quality (clique size or cut value; higher is better) and the *average wall-clock runtime per test instance* (seconds; lower is better), computed as the total runtime divided by the number of test instances. For our method, the notation $T \times K$ denotes $T$ optimization steps with $K$ particles. In particular, $T = 500$ corresponds to $\Delta t = 0.002$ with $K = 64$ particles, and $T = 100$ corresponds to $\Delta t = 0.01$ with $K = 20$ particles. A dash (—) indicates that a method is not applicable or could not be run under the given setting. The best result among learning-based methods in each column is highlighted in bold.

| **MaxCl** | | **RB-[200–300]** | | **RB-[800–1200]** | | **MaxCut** | | **BA-[200–300]** | | **BA-[800–1200]** | |
|---|---|---|---|---|---|---|---|---|---|---|---|
| **Method** | **Type** | **Size ↑** | **Time ↓** | **Size ↑** | **Time ↓** | **Method** | **Type** | **Size ↑** | **Time ↓** | **Size ↑** | **Time ↓** |
| Gurobi | OR | 19.01 | 0.230 | 33.89 | 2.360 | Gurobi | OR | 730.87 | 1.020 | 2944.38 | 9.216 |
| ERDOES | UL | 12.02 | 0.082 | 25.43 | 0.272 | ERDOES | UL | 693.45 | 0.092 | 2870.34 | 0.338 |
| LTFT | UL | 16.24 | 0.084 | 31.42 | 0.580 | LTFT | UL | 704.30 | 0.354 | 2864.61 | 2.560 |
| DiffUCO | UL | 16.22 | 0.120 | — | — | DiffUCO | UL | 727.32 | 0.120 | 2947.53 | 0.454 |
| SDDS | UL | 18.90 | 0.076 | — | — | SDDS | UL | 730.76 | 0.028 | — | — |
| RLNN | RL | 18.26 | 0.163 | 35.23 | 4.516 | RLNN | RL | 725.89 | 0.190 | 2907.18 | 0.440 |
| Greedy | H | 13.53 | 0.050 | 26.71 | 0.050 | Greedy | H | 688.31 | 0.026 | 2786.00 | 0.374 |
| MFA | H | 14.82 | 0.054 | 27.94 | 0.278 | MFA | H | 704.03 | 0.192 | 2833.86 | 0.872 |
| Ours(500×64) | UL | **19.04** | 7.071 | **40.52** | 13.28 | Ours(500×64) | UL | **731.01** | 4.162 | **2965.94** | 11.683 |
| Ours(100×20) | UL | 18.71 | 0.742 | 37.73 | 6.428 | Ours(100×20) | UL | 725.79 | 0.410 | 2960.13 | 5.622 |

*Table 3.* Comparison between Nesterov optimization and neural-guided optimization for solving Eq. (14) on MIS RB-[200–300].

| Method | Steps ↓ | Best Laplace Energy ↓ |
|---|---|---|
| NAG (w/o NN) | 500 | -15.53 |
| Ours (NN-guided) | **20** | **-19.13** |

*Figure 2.* Ablation study on MIS for **RB-[200–300]**. (a) Effect of the number of particles $K$ with $T = 20$. (b) Effect of the number of optimization steps $T$ with $K = 64$. We report the average best solution obtained within the optimization steps and the average wall-clock runtime per test instance.

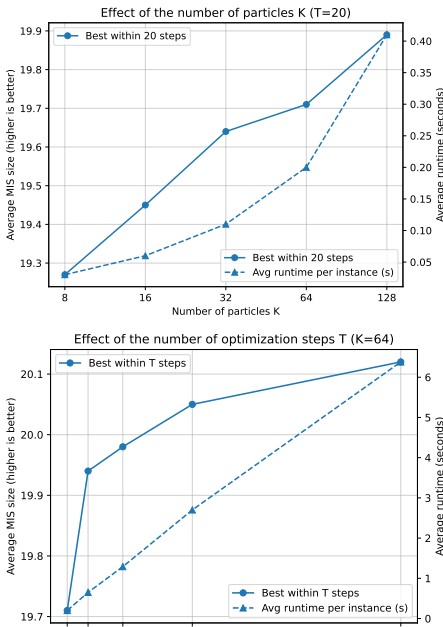

## 5.3. Ablation Study

We conduct ablation studies on the MIS benchmark **RB-[200–300]** to better understand the impact of key design choices. Unless otherwise specified, all results are averaged over the evaluation set under the same protocol as in the main experiments.

**Effect of computational budget.** We analyze how the computational budget affects solution quality and runtime by varying the number of particles $K$ and the number of optimization steps $T$ on the MIS benchmark **RB-[200–300]**. As shown in Figure 2, increasing either $K$ or $T$ consistently improves solution quality, as both lead to more extensive exploration of the solution space. However, these improvements come at the cost of increased runtime. In particular, the runtime grows approximately linearly with respect to both $K$ and $T$, while the gains in solution quality exhibit diminishing returns under larger budgets. Based on this trade-off, we adopt $K = 64$ and $T = 20$ as the default configuration in the remainder of our experiments, unless stated otherwise.

**Neural guidance vs. Nesterov optimization.** We compare our learned optimizer with a non-neural baseline that directly applies Nesterov Accelerated Gradient (NAG) method (Nesterov, 1983) to solve Eq. (14). Both methods are evaluated on MIS RB-[200–300] using the same number of particles ($K = 64$). While the NAG baseline is allowed to run for 500 optimization steps, our neural-guided method uses only 20 steps. The detailed Implementation of NAG is provided in Appendix E.3 As shown in Table 3, the neural-guided optimizer achieves better Laplace energy, demonstrating that the performance gains are not solely due to longer optimization, but stem from learning problem-aware update dynamics.

**Comparison with RLSA under equal iteration budgets.** We additionally compare our method with the recent sampling-based baseline RLSA (Feng & Yang, 2025) under strictly matched optimization budgets. Specifically, we fix the number of particles to $K = 64$ and evaluate both methods using the same number of iterations $T \in \{20, 50, 100, 200, 500\}$ on MIS RB-[200–300]. Due to space limitations, we provide the full quantitative results and detailed analysis in Appendix E.5.

## 6. Discussion and Limitations

Our framework designs objective-aware sampling dynamics for graph combinatorial optimization by combining an exact continuous multi-linear relaxation, an entropic-OT-inspired stochastic control view with a terminal potential, and an amortized GNN that approximates the induced drift function via a Laplace subproblem. While this yields a scalable learned solver in practice, the approach remains primarily empirical and does not provide global optimality guarantees; performance may vary with the training distribution and with algorithmic choices such as the number of time points/steps $T$ and the number of particles $K$. Moreover, training relies on detached rollout-based data collection and replay, so distribution shift between the replay buffer and the current policy, the Laplace approximation, and time discretization can introduce bias or additional overhead. Although the learned GNN control shows cross-scale transfer in Appendix E.4, performance can degrade when graph size or distribution differs substantially from training. Finally, the conversion from continuous samples to discrete feasible solutions is handled by greedy rounding; it works well when samples are near-binary, but tighter constraints or more fractional outputs may require stronger repair or decoding strategies.

## Impact Statement

This paper presents a learning-based optimization framework for graph combinatorial optimization. Such problems arise in a broad range of applications, including resource allocation, scheduling, routing, and network design. If deployed responsibly, improved optimization methods can help reduce computational cost and improve solution quality in operational systems, which may translate into better utilization of energy, time, and infrastructure.

At the same time, the proposed method is a general-purpose optimization technique, and its downstream impact depends on the application context. In high-stakes settings, such as logistics, infrastructure planning, or decision support, poor generalization, distribution shift, or hidden bias in training instances could lead to degraded performance or systematically suboptimal decisions. In addition, better optimization tools can also be used in applications whose broader societal effects are undesirable.

For these reasons, we view this work as a methodological contribution rather than a standalone decision-making system. Practical deployment should include application-level constraints, human oversight when appropriate, and empirical validation on the target distribution to verify reliability, robustness, and fairness.

## Acknowledgements

This work was supported in part by the Strategic Priority Research Program of Chinese Academy of Sciences (XDA0480203), the National Key R&D Program of China (No. 2025ZD0122000), the National Natural Science Foundation of China (NSFC 62273347), and the Key Research and Development Program of Jiangsu Province (BE2023016).

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

## A. Related Work

**Autoregressive models**   Autoregressive models have achieved notable success in sequence-based combinatorial optimization problems such as the **Traveling Salesman Problem** (TSP), but they exhibit certain limitations when applied to general Graph Combinatorial Optimization Problems (GCOPs). Pointer Networks (Bello et al., 2017) represent the first autoregressive framework proposed for solving TSP, where the model is trained in a supervised manner using ground-truth TSP solutions as labels. Subsequent Transformer-based models, including UTSP (Min & Gomes, 2023), NAR-CO (Wang et al., 2024), and DEITSP (Wang et al., 2025), have demonstrated strong performance by directly optimizing the objective of the continuous relaxation (GC-COP) as their training loss.

Autoregressive generation has also been adopted in Deep Generative Models (DGMs) such as GFlowNets (Bengio et al., 2023), which produce discrete samples sequentially and can therefore be applied to GCOPs (Zhang et al., 2023). Moreover, many GCOPs can be formulated as Markov Decision Processes (MDPs). For example, the process of constructing a solution to the Maximum Independent Set (MIS) problem can be modeled as an MDP, allowing reinforcement learning algorithms such as Proximal Policy Optimization (PPO) (Schulman et al., 2017; Ahn et al., 2020) to be used for optimization. Similarly, routing problems can be addressed via reinforcement learning methods like policy gradient (Kool et al., 2019).

The key difference between these problem settings lies in the nature of the input representation: routing problems are naturally expressed as ordered sequences of coordinates, whereas GCOPs such as MIS are defined on graph-structured data, which poses additional challenges for autoregressive modeling and sequential decision-making.

**Non-autoregressive models.**   The most classic non-autoregressive model for solving combinatorial optimization problems is the **Hopfield Network** (HN) (Hopfield, 1982). In recent years, works such as **Erdös Goes Neural** (Karalias & Loukas, 2020) and **QRF-GNN** (Sun et al., 2022) have employed graph neural networks (GNNs) to directly optimize the objective function of graph combinatorial optimization problems (GCOPs), achieving competitive results on tasks such as graph coloring and MaxCut. Diffusion models have also been applied to GCOPs by constructing mappings from the optimization problem to a corresponding target distribution. Two main approaches exist: one trains the diffusion model using the optimal solutions of GCOPs as labels (Sun & Yang, 2023), while the other directly treats the GCOP objective function as the loss function during training (Sanokowski et al., 2024; 2025). Moreover, dynamic programming has been creatively integrated with neural training processes to tackle GCOPs (Brusca et al., 2023).

## B. Reductions to QUBO and multilinear relaxation

### B.1. Reductions of Graph Combinatorial Optimization Problems to QUBO

In this appendix, we briefly summarize standard polynomial-time reductions from several classical graph combinatorial optimization problems to quadratic unconstrained binary optimization (QUBO). Throughout, a binary variable $x_i \in \{0,1\}$ indicates whether vertex $i$ is selected.

**Maximum Cut (MaxCut).**   Given an undirected graph with weighted adjacency matrix $\boldsymbol{W} = (w_{ij})$, the MaxCut problem seeks a partition of vertices that maximizes the total weight of edges crossing the cut. Let $x_i \in \{0,1\}$ denote the partition assignment of vertex $i$. The MaxCut objective can be written as

$$\max_{\boldsymbol{x} \in \{0,1\}^n} \sum_{i,j} w_{ij}\, x_i (1 - x_j). \tag{18}$$

Up to an additive constant, this problem is equivalent to the following QUBO formulation:

$$\min_{\boldsymbol{x} \in \{0,1\}^n} \boldsymbol{x}^\top \boldsymbol{L} \boldsymbol{x}, \tag{19}$$

where $\boldsymbol{L}$ denotes the (weighted) graph Laplacian.

**Maximum Independent Set (MIS).**   The Maximum Independent Set problem aims to find the largest subset of vertices such that no two selected vertices are adjacent. A standard QUBO formulation is

$$\min_{\boldsymbol{x} \in \{0,1\}^n} -\sum_{i=1}^n x_i + \lambda \sum_{(i,j) \in E} x_i x_j, \tag{20}$$

where $\lambda$ is a penalty parameter. For MIS, any $\lambda > 1$ makes selecting both endpoints of an edge suboptimal relative to removing one endpoint; we use $\lambda = 2$ in all experiments. Equivalently, this objective can be written in quadratic form as

$$\min_{\boldsymbol{x} \in \{0,1\}^n} \boldsymbol{x}^\top \boldsymbol{Q}_{\mathrm{MIS}} \boldsymbol{x} + \boldsymbol{c}_{\mathrm{MIS}}^\top \boldsymbol{x}, \tag{21}$$

where $\boldsymbol{Q}_{\mathrm{MIS}} \in \mathbb{R}^{n \times n}$ is defined by

$$(\boldsymbol{Q}_{\mathrm{MIS}})_{ii} = 0, \qquad (\boldsymbol{Q}_{\mathrm{MIS}})_{ij} = \begin{cases} \lambda, & (i,j) \in E, \ i \neq j, \\ 0, & \text{otherwise.} \end{cases}, \qquad (\boldsymbol{c}_{\mathrm{MIS}})_i = -1.$$

**Maximum Clique (MaxCl).** The Maximum Clique problem aims to find the largest fully connected subgraph. Let $E^c$ denote the complement of the edge set. A standard QUBO formulation is

$$\min_{\boldsymbol{x} \in \{0,1\}^n} -\sum_{i=1}^n x_i + \lambda \sum_{(i,j) \in E^c} x_i x_j. \tag{22}$$

As with MIS, $\lambda > 1$ suffices for this reduction and we set $\lambda = 2$. Equivalently,

$$\min_{\boldsymbol{x} \in \{0,1\}^n} \boldsymbol{x}^\top \boldsymbol{Q}_{\mathrm{MC}} \boldsymbol{x} + \boldsymbol{c}_{\mathrm{MC}}^\top \boldsymbol{x}, \tag{23}$$

where

$$(\boldsymbol{Q}_{\mathrm{MC}})_{ii} = 0, \qquad (\boldsymbol{Q}_{\mathrm{MC}})_{ij} = \begin{cases} \lambda, & (i,j) \in E^c, \ i \neq j, \\ 0, & \text{otherwise.} \end{cases}, \qquad (\boldsymbol{c}_{\mathrm{MC}})_i = -1.$$

## B.2. Proof of Exactness of the multilinear relaxation

*Proposition.* Let $E(\cdot; G) : \{0,1\}^n \to \mathbb{R}$ be the QUBO objective in (2), and let $f(\cdot; G) : [0,1]^n \to \mathbb{R}$ denote its multilinear extension defined in (4). Then, there exists $\boldsymbol{x}^\star \in \{0,1\}^n$ such that

$$f(\boldsymbol{x}^\star; G) = \min_{\boldsymbol{x} \in [0,1]^n} f(\boldsymbol{x}; G) = \min_{\boldsymbol{x} \in \{0,1\}^n} E(\boldsymbol{x}; G).$$

*Proof.* For any $\boldsymbol{x} \in [0,1]^n$, let $\boldsymbol{z} \sim \prod_{i=1}^n \mathrm{Bern}(x_i)$ be a random binary vector with independent coordinates. By construction of the multilinear extension,

$$f(\boldsymbol{x}; G) = \mathbb{E}\big[E(\boldsymbol{z}; G)\big].$$

Since $\{0,1\}^n \subset [0,1]^n$ and $f(\boldsymbol{x}; G) = E(\boldsymbol{x}; G)$ for all $\boldsymbol{x} \in \{0,1\}^n$,

$$\min_{\boldsymbol{x} \in [0,1]^n} f(\boldsymbol{x}; G) \leq \min_{\boldsymbol{x} \in \{0,1\}^n} E(\boldsymbol{x}; G).$$

On the other hand, for any $\boldsymbol{x} \in [0,1]^n$,

$$f(\boldsymbol{x}; G) = \mathbb{E}[E(\boldsymbol{z}; G)] \geq \min_{\boldsymbol{u} \in \{0,1\}^n} E(\boldsymbol{u}; G),$$

since the expectation of a random variable is bounded below by its minimum possible value. Taking the minimum over $\boldsymbol{x} \in [0,1]^n$ yields

$$\min_{\boldsymbol{x} \in [0,1]^n} f(\boldsymbol{x}; G) \geq \min_{\boldsymbol{u} \in \{0,1\}^n} E(\boldsymbol{u}; G),$$

which means:

$$\min_{\boldsymbol{x} \in [0,1]^n} f(\boldsymbol{x}; G) = \min_{\boldsymbol{x} \in \{0,1\}^n} E(\boldsymbol{x}; G).$$

. Let $\bar{\boldsymbol{x}} \in \arg\min_{\boldsymbol{x} \in [0,1]^n} f(\boldsymbol{x}; G)$ and define

$$m \triangleq \min_{\boldsymbol{u} \in \{0,1\}^n} E(\boldsymbol{u}; G).$$

From part (i), we have $f(\bar{\boldsymbol{x}}; G) = m$. Since

$$f(\bar{\boldsymbol{x}}; G) = \mathbb{E}[E(\boldsymbol{z}; G)], \quad E(\boldsymbol{z}; G) \geq m \text{ almost surely,}$$

the equality of the expectation implies that $E(\boldsymbol{z}; G) = m$ with positive probability. Hence, there exists at least one $\boldsymbol{u}^\star \in \{0, 1\}^n$ such that $E(\boldsymbol{u}^\star; G) = m$. Because $f(\boldsymbol{u}^\star; G) = E(\boldsymbol{u}^\star; G)$, $\boldsymbol{u}^\star$ is an integral minimizer of the continuous problem. $\qquad\square$

## C. Value Function Analysis and Proof of Proposition 4.1

We provide a self-contained derivation of the value function representation and the induced optimal drift used in Section 4. Throughout this appendix, we work on the state space $\mathcal{X} = [0, 1]^n$ and explicitly keep the graph dependence $G$ when needed.

### C.1. Stochastic Optimal Control and the HJB Equation

Consider the stochastic differential equation:

$$\mathrm{d}\boldsymbol{X}_t = \boldsymbol{u}_t(\boldsymbol{X}_t; G)\,\mathrm{d}t + \sqrt{2\varepsilon}\,\mathrm{d}\boldsymbol{W}_t, \qquad t \in [0, 1], \tag{24}$$

where $\varepsilon > 0$ is the diffusion strength and $\boldsymbol{W}_t$ is standard Brownian motion. We assume $\boldsymbol{X}_0 \sim \rho_{\mathrm{ref}}$.

Let $\Phi(\boldsymbol{x}; G)$ be a terminal potential. In our setting, $\pi^\star(\boldsymbol{x}; G) \propto \exp(-f(\boldsymbol{x}; G))$ and the terminal cross-entropy term corresponds to $\Phi(\boldsymbol{x}; G) = -\log \pi^\star(\boldsymbol{x}; G)$ up to an additive constant; we use the scaled form

$$\Phi(\boldsymbol{x}; G) = \lambda_h f(\boldsymbol{x}; G), \tag{25}$$

absorbing constants into $\Phi$.

We study the stochastic optimal control problem

$$\inf_{\boldsymbol{u}} \mathbb{E}\left[\int_0^1 \frac{1}{4\varepsilon} \|\boldsymbol{u}_t(\boldsymbol{X}_t; G)\|^2 \,\mathrm{d}t + \Phi(\boldsymbol{X}_1; G)\right]. \tag{26}$$

Define the value function

$$V(t, \boldsymbol{x}; G) = \inf_{\boldsymbol{u}} \mathbb{E}\left[\int_t^1 \frac{1}{4\varepsilon} \|\boldsymbol{u}_s(\boldsymbol{X}_s; G)\|^2 \,\mathrm{d}s + \Phi(\boldsymbol{X}_1; G) \,\Big|\, \boldsymbol{X}_t = \boldsymbol{x}\right]. \tag{27}$$

**HJB equation.**  Under standard regularity conditions, $V$ satisfies the Hamilton–Jacobi–Bellman equation

$$-\partial_t V(t, \boldsymbol{x}; G) = \inf_{\boldsymbol{u} \in \mathbb{R}^n} \left\{\frac{1}{4\varepsilon} \|\boldsymbol{u}\|^2 + \boldsymbol{u} \cdot \nabla_{\boldsymbol{x}} V(t, \boldsymbol{x}; G) + \varepsilon \Delta_{\boldsymbol{x}} V(t, \boldsymbol{x}; G)\right\}, \tag{28}$$

$$V(1, \boldsymbol{x}; G) = \Phi(\boldsymbol{x}; G). \tag{29}$$

The pointwise minimizer of the right-hand side is

$$\boldsymbol{u}_t^\star(\boldsymbol{x}; G) = -2\varepsilon \, \nabla_{\boldsymbol{x}} V(t, \boldsymbol{x}; G), \tag{30}$$

and substituting $\boldsymbol{u}^\star$ back into (28) yields the nonlinear PDE

$$\partial_t V(t, \boldsymbol{x}; G) = \varepsilon \|\nabla_{\boldsymbol{x}} V(t, \boldsymbol{x}; G)\|^2 - \varepsilon \Delta_{\boldsymbol{x}} V(t, \boldsymbol{x}; G), \qquad V(1, \boldsymbol{x}; G) = \Phi(\boldsymbol{x}; G). \tag{31}$$

### C.2. Restatement of Proposition 4.1

We restate Proposition 4.1 from the main text in a form convenient for proof.

*Proposition* (Closed-form value function and optimal drift). Let $V(t, \boldsymbol{x}; G)$ be the value function of (26). Define

$$\psi(t, \boldsymbol{x}; G) = \exp\big(-V(t, \boldsymbol{x}; G)\big). \tag{32}$$

Let the heat kernel associated with the uncontrolled diffusion $\mathrm{d}\boldsymbol{X}_t = \sqrt{2\varepsilon}\,\mathrm{d}\boldsymbol{W}_t$ be

$$p_s(\boldsymbol{x}, \boldsymbol{y}) = (4\pi\varepsilon s)^{-n/2} \exp\Big(-\frac{\|\boldsymbol{x} - \boldsymbol{y}\|^2}{4\varepsilon s}\Big). \tag{33}$$

Then the value function admits the representation

$$V(t, \boldsymbol{x}; G) = -\log\Big(\int_{\mathcal{X}} p_{1-t}(\boldsymbol{x}, \boldsymbol{y})\,\exp\big(-\Phi(\boldsymbol{y}; G)\big)\,\mathrm{d}\boldsymbol{y}\Big). \tag{34}$$

Moreover, the optimal drift satisfies

$$\boldsymbol{u}_t^\star(\boldsymbol{x}; G) = -2\varepsilon\,\nabla_{\boldsymbol{x}} V(t, \boldsymbol{x}; G) = \frac{\mathbb{E}_{\boldsymbol{Y} \sim q_{t,\boldsymbol{x}}(\cdot; G)}[\boldsymbol{Y}] - \boldsymbol{x}}{1 - t}, \tag{35}$$

where

$$q_{t,\boldsymbol{x}}(\boldsymbol{y}; G) \propto p_{1-t}(\boldsymbol{x}, \boldsymbol{y})\,\exp\big(-\Phi(\boldsymbol{y}; G)\big). \tag{36}$$

## C.3. Proof of Proposition C.2

*Proof.* We start from the nonlinear PDE (31). Apply the Hopf–Cole transform

$$\psi(t, \boldsymbol{x}; G) = \exp\big(-V(t, \boldsymbol{x}; G)\big). \tag{37}$$

A direct calculation shows that $\psi$ satisfies the linear backward heat equation

$$-\partial_t \psi(t, \boldsymbol{x}; G) = \varepsilon \Delta_{\boldsymbol{x}} \psi(t, \boldsymbol{x}; G), \qquad \psi(1, \boldsymbol{x}; G) = \exp\big(-\Phi(\boldsymbol{x}; G)\big). \tag{38}$$

By the heat kernel representation (equivalently, Feynman–Kac for this diffusion), the solution is

$$\psi(t, \boldsymbol{x}; G) = \int_{\mathcal{X}} p_{1-t}(\boldsymbol{x}, \boldsymbol{y})\,\exp\big(-\Phi(\boldsymbol{y}; G)\big)\,\mathrm{d}\boldsymbol{y}. \tag{39}$$

Taking $V = -\log\psi$ yields the closed-form representation (34).

For the optimal drift, (30) gives $\boldsymbol{u}_t^\star = -2\varepsilon\nabla_{\boldsymbol{x}} V$. Using $V = -\log\psi$, we obtain

$$\boldsymbol{u}_t^\star(\boldsymbol{x}; G) = 2\varepsilon\,\frac{\nabla_{\boldsymbol{x}}\psi(t, \boldsymbol{x}; G)}{\psi(t, \boldsymbol{x}; G)}. \tag{40}$$

Differentiate (39) under the integral sign and use $\nabla_{\boldsymbol{x}} p_s(\boldsymbol{x}, \boldsymbol{y}) = -(\boldsymbol{x} - \boldsymbol{y})p_s(\boldsymbol{x}, \boldsymbol{y})/(2\varepsilon s)$, which yields

$$\nabla_{\boldsymbol{x}}\psi(t, \boldsymbol{x}; G) = -\frac{1}{2\varepsilon(1 - t)} \int_{\mathcal{X}} (\boldsymbol{x} - \boldsymbol{y})\,p_{1-t}(\boldsymbol{x}, \boldsymbol{y})\,\exp\big(-\Phi(\boldsymbol{y}; G)\big)\,\mathrm{d}\boldsymbol{y}. \tag{41}$$

Substituting into (40) and rearranging gives

$$\boldsymbol{u}_t^\star(\boldsymbol{x}; G) = \frac{\int_{\mathcal{X}} \boldsymbol{y}\,p_{1-t}(\boldsymbol{x}, \boldsymbol{y})\,\exp\big(-\Phi(\boldsymbol{y}; G)\big)\,\mathrm{d}\boldsymbol{y}}{(1 - t)\int_{\mathcal{X}} p_{1-t}(\boldsymbol{x}, \boldsymbol{y})\,\exp\big(-\Phi(\boldsymbol{y}; G)\big)\,\mathrm{d}\boldsymbol{y}} - \frac{\boldsymbol{x}}{1 - t} = \frac{\mathbb{E}_{\boldsymbol{Y} \sim q_{t,\boldsymbol{x}}(\cdot; G)}[\boldsymbol{Y}] - \boldsymbol{x}}{1 - t}. \tag{42}$$

Recognizing the ratio as $\mathbb{E}_{\boldsymbol{Y} \sim q_{t,\boldsymbol{x}}(\cdot; G)}[\boldsymbol{Y}]$ with $q_{t,\boldsymbol{x}}$ defined in (36) yields (35), completing the proof. $\qquad\square$

## D. Laplace Approximation and Multiple-Particle Mitigation

In this section, we justify the Laplace approximation used in Section 4.4 and describe a multiple-particle strategy to mitigate the resulting approximation errors.

Recall the conditional density

$$q_{t,\boldsymbol{x}}(\boldsymbol{y}; G) \propto \exp\left(-\frac{1}{2\varepsilon}L(t, \boldsymbol{x}, \boldsymbol{y}; G)\right),$$

where $L(t, \boldsymbol{x}, \boldsymbol{y}; G)$ is defined in (13). We assume that $L(t, \boldsymbol{x}, \cdot; G)$ is lower semicontinuous on $[0, 1]^n$ and admits at least one global minimizer

$$\boldsymbol{y}^\star(t, \boldsymbol{x}; G) \in \operatorname*{argmin}_{\boldsymbol{y}\in[0,1]^n} L(t, \boldsymbol{x}, \boldsymbol{y}; G).$$

No convexity or non-degeneracy of the Hessian is required.

As $\varepsilon(1-t) \to 0$, the family of measures $\{q_{t,\boldsymbol{x}}\}_{\varepsilon>0}$ satisfies a Laplace principle, and its probability mass concentrates on the set of global minimizers of $L$ (see, e.g., Dembo & Zeitouni (1998)). In particular, for any neighborhood $U$ of the minimizer set,

$$\lim_{\varepsilon \to 0} \int_U q_{t,\boldsymbol{x};G}(\boldsymbol{y})\,\mathrm{d}\boldsymbol{y} = 1.$$

Consequently, the conditional expectation $\mathbb{E}_{\boldsymbol{Y}\sim q_{t,\boldsymbol{x}}}[\boldsymbol{Y}]$ is asymptotically supported on the minimizer set of $L$. In our algorithm, we approximate this expectation by selecting a representative minimizer $\boldsymbol{y}^\star(t, \boldsymbol{x})$, yielding

$$\mathbb{E}_{\boldsymbol{Y}\sim q_{t,\boldsymbol{x}}}[\boldsymbol{Y}] \approx \boldsymbol{y}^\star(t, \boldsymbol{x}; G), \quad \text{as } \varepsilon(1-t) \to 0,$$

which underlies the surrogate control in (15). This approximation corresponds to a first-order (MAP-level) Laplace approximation and does not rely on a second-order Gaussian expansion (see also Tierney & Kadane (1986)).

## E. Additional Experiments and Problem Reductions

### E.1. Reduction from Maximum Clique to MIS

The Maximum Clique problem seeks a largest subset of vertices such that every pair of vertices in the subset is connected by an edge. It is well known that Maximum Clique can be reduced to Maximum Independent Set (MIS) via graph complementation. Specifically, given a graph $G = (V, E)$, let $\bar{G} = (V, \bar{E})$ denote its complement graph, where $(u, v) \in \bar{E}$ if and only if $(u, v) \notin E$ for $u \neq v$. A clique in $G$ corresponds to an independent set in $\bar{G}$, and vice versa. Therefore, solving MIS on the complement graph $\bar{G}$ yields a solution to the Maximum Clique problem on $G$.

In our framework, this reduction allows clique problems to be solved by applying the proposed MIS solver directly to the complemented graph, without modifying the optimization objective or the sampling-based optimization procedure.

### E.2. Instance Generation Details

**RB Model for MIS and MaxCl**    We generate MIS instances using the RB model (Xu et al., 2005). In the RB construction, the total number of variables is given by $v = n \cdot k$, where $n$ denotes the number of groups and $k$ the number of variables per group. For small-scale instances, we sample $n \in [20, 25]$ and $k \in [5, 12]$ and retain instances with $v \in [200, 300]$. For large-scale instances, we sample $n \in [40, 55]$ and $k \in [20, 25]$ and retain instances with $v \in [800, 1200]$. The constraint density parameter $p$ is sampled uniformly from $[0.3, 1.0]$. Edges are generated according to the RB procedure, where the parameter $p$ controls the constraint density (i.e., the probability/extent of imposing constraints between variable groups), yielding instances that are typically close to the satisfiability threshold.

**Erdős–Rényi Graphs for MIS**    To complement the RB benchmarks, we additionally evaluate MIS on Erdős–Rényi (ER) random graphs. ER graphs are generated using the `nx.erdos_renyi_graph` function from NetworkX. The number of nodes is sampled uniformly from the range 700–800, and each possible edge is included independently with probability $p = 0.15$. This setting yields graphs with homogeneous random structure and moderate density, providing a generic benchmark distinct from the structured hardness of RB instances.

**BA Model for MaxCut**   For MaxCut, we generate graphs using the Barabási–Albert model as implemented in NetworkX. The number of nodes is sampled from $[200, 300]$ and $[800, 1200]$, and each new node attaches to $m = 4$ existing nodes. This setting produces scale-free networks with hub structures and heterogeneous degree distributions.

### E.3. Experimental setting

**Implementation details.**   We implement the amortized control network with a GIN backbone. For each node, the input embedding is constructed by concatenating three components: (i) the current continuous state embedding $e_c = \text{Linear}(x_c)$, (ii) a discrete sample embedding $e_d = \text{Linear}(x_d)$ where $x_d$ drawn from the Bernoulli distribution parameterized by $x_c$ (treating $x_c$ as probabilities), and (iii) a positional time embedding $e_t$ obtained from the current time $t$. We then apply a linear layer to obtain the node embedding:

$$\boldsymbol{x}_{\text{in}} = \text{Linear}\big([\boldsymbol{e}_c, \boldsymbol{e}_d, \boldsymbol{e}_t]\big). \tag{43}$$

We use a 5-layer GIN with a virtual node to facilitate long-range message passing. Each layer uses LayerNorm and residual connections:

$$\boldsymbol{x}_{k+1} = \text{LayerNorm}(\boldsymbol{x}_k + \text{GIN}(\boldsymbol{x}_k)) \tag{44}$$

We set the embedding dimension to $128$ for the large-scale datasets (suffix "800–1200" and "700–800") and to $64$ for the small-scale datasets (suffix "200–300"). We choose ReLU as activate function for all experiments.

**Training and inference setup.**   Each node input consists of three components $(x_c, x_d, x_t)$: the continuous state $x_c \in [0, 1]$, a discrete sample $x_d \in \{0, 1\}$ drawn from $\text{Bernoulli}(x_c)$ (treating $x_c$ as selection probabilities), and a time embedding $x_t$ computed from the current time $t$. At initialization, we sample the continuous states i.i.d. from $\text{Unif}[0, 1]$ and generate $x_d$ accordingly. During training, we sample times uniformly from $t \sim \text{Unif}[0, 1 - \delta]$ (with $\delta = 0.01$) to construct training pairs and populate the rollout buffer, whereas at inference time $t$ follows the deterministic discretization $t_k = k\Delta t$ with $\Delta t = (1 - \delta)/T$ as in Algorithm 2.

During training, we use $K = 8$ particles for rollout-based data collection. We set the noise level in (8) (penalty of entropy regularization term in (7)) to $\varepsilon = 0.7$ and the truncation parameter to $\delta = 0.01$ for all experiments. We use a batch size of $64$ for small-scale datasets and $16$ for large-scale datasets. The traning epoch $N_{\text{iter}}$ is set to $100$ for all datasets. The rollout length $T$ is set to $20$ points/steps (as defined in Algorithm 1) for all datasets during training process. The weighting parameter of terminal cross entropy in (9) $\lambda_h$ is set to 1 for all experiments. The penalty coefficient $\lambda$ in MIS and MaxCl is set to 2 for all experiments. We choose AdamW (Loshchilov & Hutter, 2019) as optimizer. Table 4 summarizes the main hyperparameters.

*Table 4.* Summary of main hyperparameters.

| Hyperparameter | Value |
| --- | --- |
| GNN backbone | GIN with virtual node |
| GNN layers | 5 |
| Hidden dimension | 128 (large), 64 (small) |
| Optimizer | AdamW |
| Batch size | 64 (small), 16 (large) |
| Training epochs | 100 |
| Training rollout steps | 20 |
| Entropy noise $\varepsilon$ | 0.7 |
| Truncation $\delta$ | 0.01 |
| Terminal CE weight $\lambda_h$ | 1 |
| MIS/MaxCl penalty $\lambda$ | 2 |

**Implementation details for Nesterov optimization.**   For the non-neural baseline, we solve the Laplace subproblem in Eq. (14) using Nesterov Accelerated Gradient (NAG) with a fixed learning rate $\eta = 0.1$. At iteration $k$, let $\boldsymbol{y}_k$ denote the

current iterate and $\boldsymbol{v}_k$ the momentum. We use the update

$$\tilde{\boldsymbol{y}}_k = \boldsymbol{y}_k + \beta_k \boldsymbol{v}_k, \tag{45}$$

$$\boldsymbol{g}_k = \nabla_{\boldsymbol{y}} L(t, \boldsymbol{x}, \tilde{\boldsymbol{y}}_k; G), \tag{46}$$

$$\hat{\boldsymbol{g}}_k = \boldsymbol{g}_k / \left( \|\boldsymbol{g}_k\|_2 + 10^{-8} \right), \tag{47}$$

$$\boldsymbol{y}_{k+1} = \tilde{\boldsymbol{y}}_k - \eta \, \hat{\boldsymbol{g}}_k, \tag{48}$$

$$\boldsymbol{v}_{k+1} = \boldsymbol{y}_{k+1} - \boldsymbol{y}_k, \tag{49}$$

where $\beta_k$ is the Nesterov momentum coefficient (we use the standard schedule). Since the optimization variable is constrained to $[0, 1]^n$, we apply an elementwise projection after each update:

$$\boldsymbol{y}_{k+1} \leftarrow \min\{\boldsymbol{1}, \max\{\boldsymbol{0}, \boldsymbol{y}_{k+1}\}\}. \tag{50}$$

We found that vanilla NAG is highly sensitive to step size in this setting, and the normalized-gradient update above substantially improves stability. Unless otherwise stated, we use the same penalty coefficient $\lambda = 2$ as in the MIS energy for this ablation.

### E.4. Cross-Scale Transfer

To evaluate whether the learned GNN control transfers across graph sizes, we train models on small graphs and test them directly on the corresponding large-graph benchmark using the same inference configuration ($T = 100$, $K = 20$). Table 5 compares this setting with models trained and tested on large graphs.

*Table 5.* Cross-scale transfer from small to large graphs. The ratio is relative to the large-trained model on the same test set.

| Problem | Small→Large | Large→Large | Ratio |
|---------|-------------|-------------|--------|
| MIS | 37.55 | 38.12 | 98.50% |
| MaxCl | 35.33 | 37.73 | 93.64% |
| MaxCut | 2908.46 | 2960.13 | 98.25% |

The results show moderate degradation under cross-scale transfer, suggesting that the GNN-parameterized control captures reusable graph-structured search behavior. Transfer is not lossless, especially for MaxCl, and improving robustness under larger distribution shifts remains an important direction.

### E.5. Additional Ablation Results

*Figure 3.* Comparison between our method and RLSA (Feng & Yang, 2025) on MIS RB-[200–300] under matched iteration budgets ($K = 64$, $T \in \{20, 50, 100, 200, 500\}$). We report the dataset-level average best solution quality (obj_min).

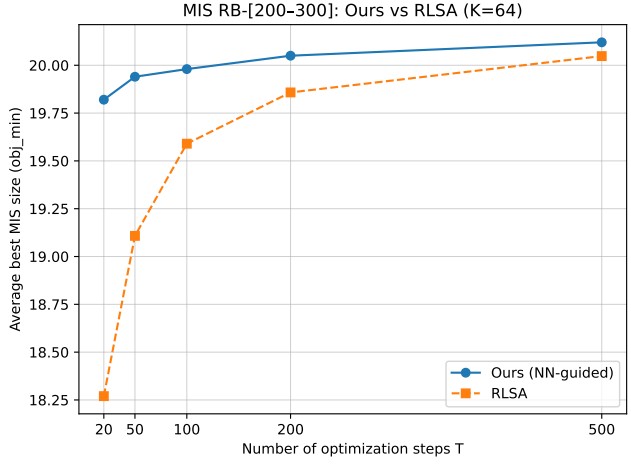

Figure 3 compares our method with the strong sampling-based baseline RLSA (Feng & Yang, 2025) under strictly matched optimization budgets. Both methods are evaluated with the same number of particles ($K = 64$) while varying the iteration budget $T \in \{20, 50, 100, 200, 500\}$.

Overall, our neural-guided solver achieves better solution quality across all budgets, with a larger gap under smaller iteration budgets. One contributing factor is the time-normalized formulation in our dynamics: our sampler is parameterized over a continuous time horizon $t \in [0, 1]$ (with truncation $\delta$), so changing $T$ primarily adjusts the discretization resolution (i.e., the step size), while the trajectory still progresses toward the terminal stage $t \approx 1$. In contrast, RLSA does not impose an explicit normalized time horizon, and small $T$ corresponds to a shorter effective evolution, which can limit exploration and progress early on. Another contributing factor is our OT-guided design principle: by encouraging distributional evolution along more direct transport paths (cf. Fig. 1), the learned dynamics can move probability mass toward high-quality regions more efficiently than hand-designed updates. Together, these effects suggest that learned, OT-informed update dynamics can substantially improve the efficiency of sampling-based optimization.

