# OpenReview forum: "Optimal Transport–Guided Stochastic Control for Graph Combinatorial Optimization"
_ICML.cc/2026/Conference — ICML 2026 regular_

### Official Review · Reviewer_MGA6 · 2026-02-18

**Soundness:** 3
**Presentation:** 3
**Significance:** 3
**Originality:** 3
**Overall Recommendation:** 5
**Confidence:** 3

**Summary:**

This paper introduces a novel approach using **diffusion samplers** to solve **quadratic unconstrained binary optimization (QUBO)** problems. The method relaxes binary decision variables to continuous ones, allowing the diffusion sampler to operate in a **constrained continuous space**. A **Laplace approximation** is employed as a loss function to learn the control, which appears to be a **novel contribution** in the context of diffusion samplers. The authors evaluate their method on a variety of popular QUBO benchmarks, demonstrating **strong performance** compared to several previous works.

**Compliance With Llm Reviewing Policy:**

Affirmed.

**Final Justification:**

I initially had some minor concerns regarding reproducibility, as certain aspects of the methodology were not sufficiently detailed in the original manuscript. These issues were addressed satisfactorily during the rebuttal phase.
I also inquired about the performance of standard diffusion samplers in this context, to which the authors responded that such approaches failed entirely. While I remain somewhat skeptical of this claim, I recognize the challenges of thoroughly investigating this matter within the limited timeframe of a rebuttal. Consequently, I did not weigh this point heavily in my assessment.
Overall, I believe the paper introduces a novel and compelling idea by leveraging a tractable Laplace approximation and the proposed algorithm demonstrates strong performance across a range of challenging combinatorial optimization benchmarks.

**Key Questions For Authors:**

## Questions
- How is the **control parameterized** in practice? For instance, do the authors use a **Langevin parameterization** (see D.4.3 in [1]), as is common in the diffusion sampler literature?
- (See also the points raised under **"Weaknesses and Clarifications Needed"**.)

[1] Vargas, Francisco, Will Grathwohl, and Arnaud Doucet. "Denoising diffusion samplers." arXiv preprint arXiv:2302.13834 (2023).

**Limitations:**

yes

**Strengths And Weaknesses:**

## Strengths
- The paper is **well-written and clearly structured**.
- The use of a **Laplace approximation as a loss function** for learning the control is innovative and, to my knowledge, unprecedented in diffusion samplers.
- The evaluation is **comprehensive**, covering a wide range of benchmarks, and the method exhibits **robust performance** across these tasks.

## Weaknesses and Clarifications Needed
1. **Experimental Details:**
   - a few details are unclear, such as the choice of **hyperparameters** and the **number of GNN layers** used. These should be explicitly stated for reproducibility.

2. **Optimization of Equation 16:**
   - The process for minimizing **Equation 16** is not clearly described. Specifically, it is unclear whether **backpropagation through \( p_t \)** is performed. Given that \( p_t \) depends on the parameters, backpropagation would seem necessary—could the authors clarify this?

3. **Boundedness of \( x \):**
   - The **SDE in Equation 8** theoretically allows \( x \) to become unbounded. However, the problem requires \( x \) to remain within the interval **[0, 1]**. How is this constraint enforced in practice?

4. **Comparison with Standard Diffusion Samplers:**
   - The authors could strengthen their analysis by comparing their approach to more conventional diffusion sampler methods, such as using **Equation 9 as a loss**. This would provide a clearer benchmark for evaluating the novelty and effectiveness of their method.

5. **Baseline Results in Table 1:**
   - It is unclear whether the baseline results reported in **Table 1** are directly taken from the original papers or if the authors reran these methods. Clarification on this point would enhance the transparency of the comparison.

6. What is H in Eq. 13?

---

> ### Author Rebuttal · Authors · 2026-03-30
>
> We thank the reviewer for their constructive comments and insightful feedback.
>
> > **W1**: hyperparameters unclear:
>
> We agree that clearer reporting of experimental details is important for reproducibility. To improve clarity, we summarize the key hyperparameters below (full details are provided in Appendix E.3):
>
> | Category       | Hyperparameter                       | Value                    |
> | -------------- | ------------------------------------ | ------------------------ |
> | Model          | GNN backbone                         | GIN (with virtual node)  |
> | Model          | # GNN layers                         | 5                        |
> | Model          | Hidden dimension                     | 128 (large) / 64 (small) |
> | Training       | Optimizer                            | AdamW                    |
> | Training       | Batch size                           | 64 (small) / 16 (large)  |
> | Training       | # Training iterations                | 100                      |
> | Training       | Rollout steps (T)                    | 20                       |
> | Regularization | Entropy noise level ($\\epsilon$)              | 0.7                      |
> | Regularization | Truncation parameter ($\\delta$)             | 0.01                     |
> | Loss           | Terminal CE weight ($\\lambda_h$)             | 1                        |
> | Task-specific  | Penalty coefficient ($\\lambda$, MIS & MaxCl) | 2                        |
>
> We will include this summary in the main text in the final version.
>
> > **W2**: ... unclear whether **backpropagation through ( p_t )** in Eq. 16
>
> We apologize for the lack of clarity. The main purpose of our objective is to guide the network to approximate the expression in Eq. 11 along trajectories that transport mass from the initial distribution toward high-probability regions of the target distribution, for which we employ a Laplace approximation. This design focuses learning on the most relevant part of the state space rather than treating all states equally.
>
> Concretely, at each training stage, we construct the training distribution using the **support of $\\rho_t$ obtained from the previous iteration**. Therefore, during the current optimization step, $\\rho_t$ is treated as a **fixed data distribution**, independent of the current network parameters. As a result, **no backpropagation is performed through $\\rho_t$**, and gradients are only computed with respect to the control network given samples from this fixed distribution. This design can be viewed as a form of **detached (or off-policy) training**, which significantly improves stability and tractability.
>
> > **W3**: Boundedness problem:
>
> The GNN outputs are passed through a **sigmoid function**, ensuring that the predicted states lie in $[0,1]$. In addition, after injecting stochastic noise, we apply a **clamping (projection) step** that truncates values outside the interval back to $[0,1]$.
>
> This combination ensures that all states remain feasible throughout both training and inference. We will clarify this implementation detail in the revised version.
>
> > **W4**: Comparison with Standard Diffusion Samplers:
>
> In our setting, directly using Equation 9 as a loss leads to a **degenerate solution**, where the learned control collapses to $u_t = 0$ for almost all $t \\in [0,1)$, and only adjusts samples near the terminal time, where the cross-entropy loss is applied. We hypothesize that this trivialization arises due to a combination of factors, including the loss weighting, the network architecture, the parameterization of the control, and the problem formulation itself. In contrast, our approach in Equation 16 avoids this issue by providing supervision across intermediate timesteps, resulting in non-trivial learned control dynamics. We will clarify this distinction in the revised manuscript.
>
> > **W5&W6**:  Baseline Results in Table 1 and  H in Eq. 13:
>
> We apologize for the confusion. The symbol $H$ used in Algorithm 2 and Eq. 13 is a typographical inconsistency. The correct notation is $\\Phi(y; G)$, as formally defined in Proposition 4.1. About baseline results, for PPO and RLNN, we re-ran the experiments. For the remaining baselines, we report results from the original papers. We will explicitly clarify this in the final version for transparency.
>
> > **Q1**: How is the **control parameterized** in practice ... :
>
> We parameterize the control velocity field $u_t(x;G)$ **directly using a GNN**, with a final sigmoid output to ensure boundedness. Unlike standard diffusion sampler that combines two neural networks with score of target distribution, our approach uses a single GNN to learn the control. We speculate that this difference in parameterization may be one contributing factor to the trivialization in **Q4** of the control when using Equation 9 as a loss (i.e., $u_t \\approx 0$ for most $t$ except near the terminal time).

---

> > ### Author Rebuttal · Reviewer_MGA6 · 2026-04-01
> >
> > I thank the authors for their answer and my questions have been adressed and I recommend to accept this paper.

---

> > > ### Author Response · Authors · 2026-04-01
> > >
> > > Thank you very much for your positive feedback and for your support of our work.
> > > We are glad that our responses have addressed your concerns, and we sincerely appreciate your recommendation.

---

### Official Review · Reviewer_nWQF · 2026-03-12

**Soundness:** 2
**Presentation:** 2
**Significance:** 2
**Originality:** 3
**Overall Recommendation:** 4
**Confidence:** 2

**Summary:**

This paper studies graph combinatorial optimization. The main idea is to motivate the sampling dynamics using entropic OT / Schrödinger bridge ideas. I was not fully convinced that the OT component is presented clearly enough or that its role is sufficiently isolated from the other parts of the method.

**Compliance With Llm Reviewing Policy:**

Affirmed.

**Final Justification:**

I thank the reviewers for addressing my questions. All questions are well addressed, and I agree with some of the other reviewers' comments regarding the new technicality. Based on this, let me raise my score to 4.

**Key Questions For Authors:**

1. What exactly does OT-guided mean here? Is OT mainly a conceptual motivation, or does the learned dynamics approximate an entropic OT / Schrödinger bridge object in a precise sense?
2. If one has access to a parameterized transport map between the reference and target distributions, can it be incorporated directly into your framework? If not, what is the main advantage of the stochastic-control formulation?
3. What’s the theoretical novelty in Proposition 4.1?

**Limitations:**

Yes

**Strengths And Weaknesses:**

The paper proposes an interesting framework combining energy-based sampling and graph-aware learning. My main concern is clarity. The paper includes many ingredients—relaxation, Boltzmann sampling, OT, stochastic control, Laplace approximation, and GNN amortization—so it is hard to identify the core contribution. In particular, the meaning of “OT-guided” remains somewhat vague: OT seems to be more of a motivating principle than a clearly preserved object in the final algorithm.

---

> ### Author Rebuttal · Authors · 2026-03-30
>
> We thank the reviewer for their careful reading and for highlighting points in the paper.
>
> > **W1&Q1&Q2**: OT-guided meaning... If one has access to a parameterized transport map ... :
>
> We apologize for the lack of clarity. In our framework, **OT-guided** refers to the fact that the optimization dynamics are formulated through a stochastic control problem that closely follows the structure of entropic optimal transport / Schrödinger bridge.
>
> Specifically, we introduce a controlled diffusion process (Eq. 8), where a time-dependent control $u_t$ transports mass from a reference distribution toward a target distribution. The objective in Eq. (9) includes a quadratic control cost $\\int_0^1 \\mathbb{E}\\|u_t(X_t)\\|^2 dt$, which is characteristic of entropic OT / Schrödinger bridge formulations. In particular, this formulation can be viewed as the **dynamic (stochastic control) formulation of entropic OT / Schrödinger bridge** [1].
>
> At the same time, our method departs from standard entropic OT / Schrödinger bridge in the **terminal objective**. We use a cross-entropy term $\\mathrm{CE}[\\rho_1 \\| \\pi^* ] $, which encourages the terminal distribution to place more mass on high-probability (low-energy) regions of $\\pi^*$. As a result, the learned dynamics guide samples toward lower-energy regions of the energy landscape, which is aligned with our goal of solving optimization problems.
>
> Therefore, OT is not merely a high-level motivation: it **determines the structure of the stochastic control formulation (Eq. 8–9)**, while the modified terminal objective adapts this framework to optimization, leading to a different solution behavior from standard OT/SB.
>
> Regarding a parameterized transport map, if such a map were available, it could in principle provide useful information. However, its relationship to our formulation is **not direct**, because our framework is built on controlled stochastic dynamics with noise, state constraints on $[0, 1] $, and a terminal objective that is not the standard OT/SB matching objective. For this reason, it is unclear whether a transport map can be incorporated in a principled way without modifying the formulation.
>
> The stochastic control formulation offers several advantages for optimization:
>
> 1. The evolution dynamics (Eq. 8) include noise, which allows the process to escape local optima while the terminal cross-entropy encourage to the mode of target distribution.
> 2. The formulation admits mature solution techniques from stochastic control, such as log-transformations and Feynman–Kac representations.
> 3. This form is particularly amenable to neural network parameterization and optimization, allowing scalable approximation of the optimal dynamics.
>
> We will revise the manuscript to clarify this interpretation, better highlight the advantages of our framework, and make this distinction more explicit in the presentation.
>
> [1] Zhang, Q et al. Path Integral Sampler: a stochastic control approach for sampling. arXiv preprint arXiv:2111.15141. (2021).
>
> >**W1**: hard to identify the core contribution
>
> Our core contribution is to develop a **learnable stochastic-control-based framework, inspired by optimal transport, for solving graph combinatorial optimization problems**.
>
> Specifically:
> (1) we leverage the equivalence between the continuous multilinear relaxation and the original discrete problem to obtain a tractable continuous formulation;
> (2) we construct a stochastic control problem (Eq. 9), inspired by optimal transport, to characterize the solution;
> (3) we parameterize the resulting control using a GNN together with a Laplace approximation for scalable approximation.
>
> In this view, relaxation, OT, stochastic control, and GNNs are not independent components, but serve as building blocks of a unified framework centered on this stochastic-control formulation.
>
> > **Q3**: novelty in Proposition 4.1:
>
> We note that, for readers familiar with stochastic optimal control, the representation in Proposition 4.1 can be seen as an application of classical HJB theory and log-transformations to our specific problem setting.
>
> The purpose of Proposition 4.1 is instead to make explicit a particular structure that is crucial for our method. Specifically, it shows that the optimal control can be written as a conditional expectation (Eq. 11), which directly motivates our algorithmic design.
>
> Importantly, this representation is closely tied to our choice of terminal objective. The cross-entropy structure enables a tractable form of the value function, which in turn leads to the conditional expectation representation in Eq. (11).
>
> To the best of our knowledge, this perspective has not been explicitly used to derive learnable optimization dynamics for combinatorial problems.

---

> > ### Author Rebuttal · Reviewer_nWQF · 2026-04-04
> >
> > Thank you to the authors for their response. I have read the rebuttal carefully, but the significance of this work is still not entirely clear to me. In particular, the connections among stochastic optimal control, entropic optimal transport, and the Schrödinger bridge are already well studied. While I agree with the authors that the perspective in Proposition 4.1 appears to be new in this context, I am not yet convinced that this is sufficient to change my overall assessment. Therefore, I would like to maintain my score.

---

> > > ### Author Response · Authors · 2026-04-04
> > >
> > > Thank you for your thoughtful feedback and for carefully considering our rebuttal.
> > >
> > > We would like to further clarify that our goal is not to introduce a new connection among stochastic optimal control, entropic optimal transport, and the Schrödinger bridge, as these relationships are well established. **Instead, our aim is to leverage this perspective to formulate and solve graph combinatorial optimization problems within a stochastic control framework.** When the terminal objective is defined via KL divergence, stochastic optimal control is equivalent to EOT/SB under certain conditions [1]. However, directly adopting a KL-based terminal objective presents several challenges in our setting:
> > >
> > > 1. The resulting optimal control $u\_t$ typically involves entropy terms of the terminal distribution, which itself depends on the control $u\_t$, leading to a less explicit and harder-to-handle formulation;
> > > 2. These entropy-related terms introduce additional complexity in approximation, especially when parameterizing the control with neural networks (we need Hutchinson estimator to estimate entropy);
> > > 3. Such formulations are primarily designed for distribution matching and do not explicitly encourage concentration on low-energy (high-quality) solutions, which is less aligned with the goal of combinatorial optimization.
> > >
> > > In contrast, adopting a cross-entropy terminal objective leads to the following advantages:
> > >
> > > 1. The resulting control exhibits a more explicit dependence on the energy (objective) function, yielding a cleaner and more interpretable form;
> > > 2. The induced structure is more amenable to learning and approximation, reducing computational complexity in practice;
> > > 3. The resulting dynamics more directly concentrate samples toward high-quality (low-energy) solutions, which is better aligned with optimization objectives.
> > >
> > > Therefore, we view the resulting formulation with a cross-entropy terminal objective as distinct from the classical EOT/SB framework, particularly in that it does not correspond to a standard distribution-matching objective between simple distributions $q(x)$ and $\\pi^*(x)\sim \\exp(-f(x))$.
> > >
> > > We hope this clarification better highlights the significance of our contribution.
> > >
> > > [1] Zhang, Q et al. Path Integral Sampler: a stochastic control approach for sampling. arXiv preprint arXiv:2111.15141. (2021).

---

### Official Review · Reviewer_hM7w · 2026-03-12

**Soundness:** 2
**Presentation:** 2
**Significance:** 1
**Originality:** 2
**Overall Recommendation:** 3
**Confidence:** 3

**Summary:**

This paper proposes an Optimal Transport (OT)-guided stochastic control framework to solve graph Combinatorial Optimization (CO) problems. The authors first take a Quadratic Unconstrained Binary Optimization (QUBO) formulation and apply an exact multilinear continuous relaxation. Recognizing that this resulting continuous energy landscape is highly non-convex, they treat it as an energy function and attempt to sample from its induced Boltzmann distribution to avoid poor local optima. By framing this sampling process as an optimal transport problem, moving from a simple reference distribution to the target Boltzmann distribution, they derive a stochastic optimal control problem. Finally, they parameterize the control policy using Graph Neural Networks (GNNs) to simulate the optimal sampling dynamics.

**Compliance With Llm Reviewing Policy:**

Affirmed.

**Key Questions For Authors:**

- How does the numerical solver handle the massive gradient instabilities introduced when heavily constrained problems are reduced to QUBO via large penalty terms?

- Stochastic optimal control requires simulating continuous trajectories, meaning the GNN must be evaluated at multiple time steps for multiple particles. How does the actual wall-clock inference time of your method compare to highly-optimized discrete baselines when scaling to dense graphs with thousands of nodes?

- How does the proposed OT-guided control compare empirically against recent continuous samplers for CO, such as RLSA (Feng & Yang)? Could authors also compare with other methods like discete diffusions or GFlownets?

**Limitations:**

See weakness section.

**Strengths And Weaknesses:**

**Strength**

- The paper is well-structured and provides an elegant conceptual perspective. Framing the sampling of the induced Boltzmann distribution as an optimal transport and probability mass flow problem provides a highly intuitive geometric interpretation of the sampling dynamics.

- The authors provide a solid empirical evaluation on standard benchmark tasks, demonstrating competitive results against the selected baselines.

**Weakness**

- The core novelty of the paper is somewhat limited, as it primarily combines two well-established concepts: (1) the exact multilinear continuous relaxation of pseudo-Boolean/QUBO functions, and (2) applying neural optimal transport/stochastic control to sample from unnormalized Boltzmann distributions (standard in recent generative modeling literature). Moreover, the surrogate loss functional in Equation (13) is a highly standard proximal operator objective, widely known and utilized in literature.

- A major shortcoming of this application is its strict reliance on the QUBO formulation. Real-world combinatorial optimization problems often involve complex, higher-order, or strict inequality constraints (e.g., routing windows, capacity limits). While the authors note that constrained problems like MIS can be reduced to QUBO, doing so requires introducing massive penalty scalars.

- The experimental section lacks comparisons to recent comparisons such as RSLA (Feng, S. and Yang, Y. Regularized langevin dynamics for combinatorial optimization).

- The presentation suffers from a confusing notational leap regarding the energy function. The variable $H(\cdot; G)$ is abruptly introduced in Algorithm 2 and Equation 13 without a clear, immediate definition in the text.

---

> ### Author Rebuttal · Authors · 2026-03-30
>
> We thank the reviewer for their constructive comments and valuable feedback, which have helped improve the clarity of our work.
>
> ---
>
> > **W1**: combines two well-established concepts :
>
> We would like to clarify that the contribution goes beyond a simple combination of two standard ingredients. First, while exact multilinear relaxations of pseudo-Boolean/QUBO functions are known, prior work typically uses them as continuous surrogates for optimization. Our key step is different: we exploit their exact correspondence with the original discrete objective to **reformulate graph combinatorial optimization as a stochastic control problem over the induced energy landscape** (Eq. 9).
>
> Second, although neural OT / stochastic control methods are known in generative modeling, our objective and derivation are different. The terminal cross-entropy objective encourages the terminal distribution to concentrate on low-energy (high-probability) regions of the energy landscape, making the resulting dynamics more concentrated on such regions than standard generative distribution-matching formulations, in a way that is aligned with our goal of solving optimization problems. Under this objective, the value function admits the representation in Eq. 11, which directly leads to the training objective in Eq. 13. Thus, Eq. 13 is not introduced as a generic proximal heuristic; although it resembles a proximal objective in form, it is **derived from the stochastic control formulation under our terminal objective**.
>
> Overall, while the individual components connect to known literature, the novelty lies in the non-trivial reformulation and derivation that unify exact multilinear relaxation, OT-guided stochastic control, and amortized GNN-based inference into one framework for graph CO.
>
> ---
>
> > **W2&Q1**: ... strict reliance on the QUBO ... massive penalty scalars ... massive gradient instabilities ... :
>
> We note that QUBO/Ising provides a standard and expressive modeling interface for a broad class of combinatorial optimization problems, as discussed in [1] and in Section 2.2 of [2], including problems with linear constraints such as routing windows and capacity limits. QUBO uses binary variables in $\\{0,1\\}$, while the Ising formulation uses spin variables in $\\{-1,1\\}$, with equivalent representations up to a change of variables. Through standard reductions, both constrained and higher-order formulations can be encoded in this framework.
>
> In our work, QUBO serves as a **convenient entry point**, which enables the stochastic-control formulation. At the same time, the framework developed from Section 4 onward is **not tied to QUBO itself**: once the problem is expressed as a continuous non-convex energy, the same methodology applies more generally.
>
> For **large penalty terms**, our statement in the paper (“$\\lambda > 0$ is chosen sufficiently large”) can be made more precise. For MIS, a threshold such as $\\lambda > 1$ already suffices to preserve correctness, and we use $\\lambda = 2$ in experiments. More broadly, [1] also discusses penalty constructions for wider classes of problems, where suitable penalty values preserve equivalence between the original problem and its QUBO representation.
>
> Regarding **gradient stability**, we did not observe instability in practice. The only gradients are computed during network training, where standard stabilization techniques such as gradient clipping are applied. We agree that broader empirical evaluation on more general constrained settings would strengthen the paper, and we will clarify this scope in the revision.
>
> [1] Andrew Lucas. Ising formulations of many NP problems. arXiv:1302.5843 (2013).
> [2] Glover et al. The Quadratic Unconstrained Binary Optimization Problem (2022).
>
> ---
>
> >**W4**: variable is abruptly introduced :
>
> We apologize for the confusion. The variable should be $\Phi(\boldsymbol{y};G)$, as defined in Proposition 4.1. We will correct this in the revision.
>
> ---
>
> > **W3&Q2&Q3**: ...highly-optimized discrete baselines...RLSA... diffusions or GFlownets... :
>
> These comparisons are already included in Tables 1 and 2 of the main paper.
>
> **Highly-optimized discrete solvers.** We compare against **Gurobi** and **KaMIS**. On large MIS (RB-[800–1200]), our solution quality (39.25) is below Gurobi (40.91) and KaMIS (43.15), but the **wall-clock runtime per graph is faster** (ours: 9.93s; Gurobi: 15.62s; KaMIS: 14.76s). On MaxCl and MaxCut, we obtain improved solution quality (e.g., 40.52 vs. 33.89; 2965.94 vs. 2944.38) at moderately higher runtime.
>
> **Learning-based baselines.** We compare against **RLSA (RLNN)**, **LTFT**, and **DiffUCO**. Our method consistently improves solution quality across tasks; for example, on MIS RB-[800–1200], we obtain 39.25 vs. 38.06 (RLNN) and 38.87 (DiffUCO), and on MaxCut BA-[800–1200], 2965.94 vs. 2907.18 (RLNN) and 2947.53 (DiffUCO). The trade-off is moderately higher runtime due to trajectory simulation.

---

> > ### Author Rebuttal · Reviewer_hM7w · 2026-04-04
> >
> > I remain unconvinced by the overall algorithmic formulation, mainly because the method appears to rely on multiple layers of approximation, each of which significantly changes the original problem. Please correct me if I misunderstood something.
> >
> > 1. **Discrete-to-continuous relaxation.**
> >    The starting point is a discrete quadratic combinatorial optimization problem, but the method immediately applies a multilinear relaxation and replaces the original discrete state space with a continuous domain $x \in [0,1]^n$. This is already a strong approximation, since the problem is no longer solved in its native discrete space.
> >
> > 2. **Relaxation in the target matching / terminal condition.**
> >    The paper formulates an entropic OT problem from a predefined prior distribution (e.g., a standard Gaussian) to the target distribution $\pi^\star$. However, even this formulation does not appear to match the target exactly, since the terminal condition is further relaxed through a cross-entropy cost (Eq. 19). This introduces another approximation and effectively replaces the original objective with a surrogate control problem.
> >
> > 3. **Approximation in the optimal control computation.**
> >    While the expression for the optimal control $u^\star$ in Eq. 11 is standard from the HJB / tilted matching perspective, the practical computation of this control requires yet another substantial approximation through a simplified treatment of the Feynman–Kac representation. In my opinion, this is not a small implementation detail, but a major approximation at the core of the method.
> >
> > Taken together, these steps make the final algorithm feel far from the original problem. In particular, the approximations in the discrete-to-continuous relaxation and in the control computation seem especially strong, and at present I do not see sufficient evidence that they preserve the essential structure of the original optimization problem.
> >
> > 4. **Need for parameter ablations.**
> >    My understanding is that these approximations are only justified in certain asymptotic regimes, for example when $\lambda$ is very large (like 100-1000) and $\epsilon$ is very small (like 1e-3). If so, the paper should provide explicit ablations over these parameters to demonstrate when the approximations are accurate and when they fail. Without such evidence, it is difficult to assess the validity of the proposed method.
> >
> > 5. **Unclear motivation for the entropic OT formulation.**
> >    I also share the concern that the motivation for introducing the entropic OT formulation is not sufficiently convincing. There are already discrete solvers that do not require these relaxations, such as MDNS. In addition, once the problem has already been relaxed to a continuous state space, it is unclear why one should further formulate it as an entropic OT / stochastic control problem, instead of directly applying continuous diffusion-based samplers such as AM, PDDS, PIS, or DDS. Moreover, even with entropic OT formulation, we can apply recent SB sampler ASBS. At present, the method seems to convert the original problem into a more complicated surrogate, and then rely on several approximations to make that surrogate tractable.
> >
> > Overall, I do not yet see a compelling reason why this specific chain of approximations is necessary. There seem to be several alternative design choices that could avoid some of these approximations altogether. Since prior work has already considered the discrete-to-continuous relaxation itself, the authors should more clearly justify why the additional entropic OT and control-based approximations are needed, and what concrete advantages they provide over simpler alternatives.
> >
> > **References**
> >
> > MDNS: Masked Diffusion Neural Sampler
> >
> > AM: Adjoint Matching
> >
> > PDDS: Particle Denoising Diffusion Sampler
> >
> > PIS: Path Integral Sampler
> >
> > DDS: Denoising Diffusion Sampler
> >
> > ASBS: Adjoint Schrodinger Bridge Sampler

---

> > > ### Author Response · Authors · 2026-04-05
> > >
> > > Thank you for the careful reading and thoughtful questions. We believe the concern stems from interpreting our method as a sequence of independent approximations. In contrast, our framework should be understood as a **single reformulation** of the original combinatorial optimization problem into a stochastic control problem, where each component plays a distinct role rather than introducing arbitrary approximations.
> > >
> > > ### 1. **Discrete-to-continuous relaxation.**
> > >
> > > For the discrete-to-continuous relaxation, we emphasize that this step **does not alter the set of global optima**. Due to the multilinear structure (Proposition 3.1 & Appendix B.2), the relaxed problem over $[0,1]^n$ still admits discrete optimal solutions that coincide with those of the original problem. Therefore, this step should not be viewed as moving away from the original problem.
> > >
> > > ------
> > >
> > > ### 2. **Relaxation in the target matching / terminal condition.**
> > >
> > > The terminal objective is designed to **prioritize concentration on high-probability (low-energy) regions** of $\\pi^\\star$, rather than matching the full distribution. This design reflects the optimization setting: the goal is not to faithfully reproduce $\\pi^\\star$, but to **identify high-quality solutions**. In this sense, the cross-entropy objective is not an arbitrary relaxation, but a deliberate choice that aligns the dynamics with optimization rather than distribution matching. We will revise the manuscript to clarify this distinction more explicitly.
> > >
> > > ------
> > >
> > > ### 3. **Approximation in the optimal control computation.**
> > >
> > > We agree that the control approximation becomes more accurate in asymptotic regimes such as $\\epsilon(1-t)\\to 0$( Appendix D ).  On the one hand, $\\epsilon$ controls the noise level. While smaller $\\epsilon$ improves approximation accuracy, it reduces the ability to escape poor local optima. Empirically, we observe degraded performance when $\\epsilon$ is too small:
> > >
> > > | MIS(RB-200-300)\noise level $\\epsilon$ | 0.9   | 0.7   | 0.5   | 0.3   | 0.1   | 0.01  | 0.001 |
> > > | -------------------------------------- | ----- | ----- | ----- | ----- | ----- | ----- | ----- |
> > > | **Results** ($\\uparrow$)               | 19.55 | 19.65 | 19.61 | 19.66 | 19.59 | 18.95 | 18.49 |
> > >
> > > This illustrates a trade-off: moderate noise improves performance by enabling exploration, while overly small $\epsilon$ leads to degradation despite better approximation. On the other hand, time discretization affects how well the dynamics approximate the terminal behavior. Finer discretization near $t=1$ (smaller $\\Delta t$) improves performance, consistent with the ablation in Fig. 2.
> > >
> > > ---
> > >
> > > ### **Summary of the above points**
> > >
> > > Taken together, these components should not be viewed as a chain of approximations that progressively move away from the original problem. Instead, they form a **unified reformulation**: the multilinear relaxation preserves the optimization structure, the stochastic-control formulation provides a principled way to navigate the energy landscape, and the terminal objective and control approximation make the problem computationally tractable while remaining aligned with optimization.
> > >
> > > ---
> > >
> > > > **Unclear motivation for the entropic OT formulation ... concrete advantages they provide over simpler alternatives.**
> > >
> > > Given that the relaxation preserves the original objective structure, our goal becomes to efficiently approach high-quality solutions of the continuous objective $f$.
> > >
> > > We adopt an entropic OT formulation for this purpose: the OT component provides a principled way to transport mass toward the high-probability (low-energy) regions of $\\pi^\\star \\propto \\exp(-f)$, while the entropic term (controlled by $\\epsilon$) introduces stochasticity that helps avoid poor local optima. In particular, when $\\epsilon \\to 0$, the formulation approaches deterministic optimal transport, while positive $\\epsilon$ introduces stochasticity that facilitates exploration. This naturally leads to a stochastic-control formulation, whose value function representation (Eq. 11) provides a direct link between the energy function and the dynamics. To make the computation tractable, we further introduce approximations in the control, resulting in a practical training objective.
> > >
> > > Compared with methods primarily designed for distribution matching (e.g., MDNS, AM, PDDS), our formulation places stronger emphasis on concentrating samples in low-energy regions, which is more aligned with the optimization goal. This is also consistent with the empirical advantage observed over discrete diffusion-based samplers such as DiffUCO (Table 1), which are specifically designed for graph combinatorial optimization. This highlights the **key advantage** of our formulation in the optimization setting.
> > >
> > > We acknowledge that this motivation was not clearly explained in the original manuscript and will clarify it in the revision.

---

### Official Review · Reviewer_7ZZa · 2026-03-17

**Soundness:** 4
**Presentation:** 3
**Significance:** 3
**Originality:** 3
**Overall Recommendation:** 5
**Confidence:** 3

**Summary:**

This paper considers graph combinatorial optimization (GCO) problems and proposes a stochastic control framework based on optimal transport (OT) to solve them. The paper relies on continuous bilinear relaxations of the GCO problems, reformulated as quadratic binary optimization problems, which are shown to be exact. To address the highly non-convex resulting optimization landscape, an OT-guided sampling approach is proposed, in which a reference distribution is efficiently transported to a target distribution defined by the resulting objective energy landscape. To address the intractability of the stochastic optimal control problem, a graph neural network (GNN) is trained to minimize the Laplace energy using rollout samples collected in a replay buffer, and the trained GNN's outputs are then used to perform inference and obtain a tractable approximate control. Numerical results on various graph types and sizes in three prominent GCO problems (Maximum Independent Set, Maximum Cut, and Maximum Clique) highlight the efficacy of the proposed method, especially as compared to traditional optimization-based and prior learning-based approaches in the literature.

**Compliance With Llm Reviewing Policy:**

Affirmed.

**Final Justification:**

Given the paper's contribution to the field of graph combinatorial optimization, its high writing quality, and the authors' response to all reviewers' comments, I recommend acceptance.

**Key Questions For Authors:**

1. Is there a reason to use the term "multi-linear" instead of "bilinear" for the optimization problem relaxation throughout the paper? Are there instances of graph problems where the problem formulation is not bilinear?
2. Have you tried any alternatives to the greedy approach mentioned at the end of Section 4.4? This last step seems crucial to the success of the final result, and I wonder if it can be improved further.
3. I believe there is no reference to Algorithm 2 in the text. Please refer to it and potentially swap its order with Algorithm 1, since the GNN should be trained before inference can be run.
4. Are graph size ranges fixed during training vs. inference? Given that GNNs are size- and permutation-invariant, have you tried training on smaller graphs and transferring the trained GNN to larger graphs during inference?

**Limitations:**

More extended discussion on where the method fails could be an interesting and important addition to Sections 5 or 6.

**Strengths And Weaknesses:**

**Strengths**
- The paper is very well written, accessible, and easy to follow.
- The ingredients of the proposed solution (bilinear relaxation, stochastic control, optimal transport, and graph neural networks) are all combined synergistically to introduce an interesting method for graph combinatorial optimization problems.
- The proposed method delivers promising performance in a wide range of experimental settings.


**Weakness**
- The optimality gap of the proposed method is unclear. I would have liked a more extensive discussion of whether such gaps could be proved theoretically for a subclass of problems, or, at the very least, some failure cases that highlight where the method falls short.

---

> ### Author Rebuttal · Authors · 2026-03-30
>
> We thank the reviewer for the positive evaluation and constructive feedback.
>
> > **W1**: optimality gap of the proposed method is unclear
>
>  A precise characterization of the optimality gap in our framework is inherently challenging and beyond the scope of this work, due to the combination of stochastic dynamics, sampling, and neural approximation. The gap depends on several sources of approximation, including: (i) the alignment between the terminal distribution induced by the stochastic control problem (Eq. 9) and the target energy function, (ii) sampling error from finite samples, (iii) discretization error due to finite time steps, and (iv) approximation and generalization error of the neural parameterization.
>
> While a full theoretical bound is beyond the scope of this work, these sources of error are well-understood and can be controlled in principle (e.g., via increased sample size and finer time discretization). Identifying subclasses of problems for which such guarantees can be established remains an open question. We expect larger gaps in settings where the energy landscape is highly irregular or strongly multi-modal, which makes both sampling and optimization more challenging. A systematic characterization of such cases remains an interesting direction for future work.
>
> > **Q1**: term "multi-linear" instead of "bilinear" ... multi-linear graph CO:
>
> In our current formulation and experiments, the objective is indeed bilinear. We use the term “multilinear” in a broader sense to reflect the general polynomial structure of the relaxation, rather than to imply the presence of higher-order interactions in our current implementation.
>
> That said, certain combinatorial problems (e.g., Minimum Dominating Set) can naturally lead to higher-order terms before reduction, although such formulations can typically be converted to bilinear ones via auxiliary variables.
>
> > **Q2**: ...greedy approach at the end of Section 4.4:
>
> We thank the reviewer for the suggestions regarding the rounding step. In our setting, the learned solutions are typically already very close to binary (i.e., values concentrate near 0 or 1), which is consistent with the properties of multilinear relaxations. As a result, the final rounding step has a relatively limited impact on performance.
>
> We have explored simple alternatives to the greedy approach, such as thresholding at 0.5 followed by minor corrections to enforce feasibility when constraints are violated (which occurs rarely in practice). This approach yields slightly worse performance, but remains competitive.
>
> We also considered more advanced decoding strategies such as Cross-Entropy (CE) or Token-CE methods [1], which are commonly used for rounding. However, since our solutions are already near-binary, these methods provide limited additional improvement.
>
> We agree that the final rounding step is important for obtaining feasible discrete solutions. In our setting, however, the learned continuous solution is already close to binary, which makes a simple greedy procedure (e.g., selecting a maximal independent set) sufficient to recover high-quality discrete solutions.
>
> [1] Sebastian et al., A Diffusion Model Framework for Unsupervised Neural Combinatorial Optimization. arXiv preprint arXiv:2406.01661 (2024).
>
> > **Q3**: no reference to Algorithm 2 in the text:
>
> We agree that the reference to Algorithm 2 is missing in the text, and that the current ordering may cause confusion. This is an issue in presentation. We will revise the manuscript to explicitly refer to Algorithm 2 in the appropriate context and reorder the algorithms to ensure that the training procedure is presented before the inference stage.
>
> > ***Q4***: graph size ranges fixed during training vs. inference:
>
> In our current experiments, the graph sizes are fixed during both training and inference for each problem. We have not previously evaluated cross-scale transfer.
>
> To address this, we conducted additional experiments where the GNN was trained on smaller graphs and tested on larger graphs for each combinatorial optimization problem. For fair comparison, all experiments use the same inference configuration (i.e., $T=100, K=20$).
>
> The results are summarized below:
>
> | Problem | Train (small) → Test (large) | Train (large) → Test (large) | Ratio (%) |
> | ------- | ---------------------------- | ---------------------------- | --------- |
> | MIS     | 37.55                        | 38.12                        | 98.50%    |
> | MaxCl   | 35.33                        | 37.73                        | 93.64%    |
> | MaxCut  | 2908.46                      | 2960.13                      | 98.25%    |
>
> The ratio is computed relative to the performance of models trained and tested on large graphs.
>
> These results indicate that our GNN-based control can generalize reasonably well across graph sizes, with only moderate performance degradation. We will include these additional results and discussion in the revised manuscript.

---

> > ### Author Rebuttal · Reviewer_7ZZa · 2026-03-31
> >
> > Thank you for your rebuttal. I maintain my favorable rating for the paper, and I think it presents an interesting contribution.

---

> > > ### Author Response · Authors · 2026-04-01
> > >
> > > Thank you for your positive feedback and for taking the time to review our rebuttal.
> > > We are glad that our clarifications have addressed your concerns, and we appreciate your support and encouraging comments.

---

### Decision · Program_Chairs · 2026-04-30

**Decision:**

Accept (regular)

**Comment:**

This paper proposes an OT-guided sampling framework for graph combinatorial optimization. The overall strength and weakness of the paper are as follows.

Strength
- The paper is well-written.
- The proposed solution demonstrates good performance in the experiments.

Weakness
- More related baselines could be compared in the experiments.

Overall, the reviewers find the paper to be interesting and most concerns were addressed during the rebuttal.